# Decision-making strategies implemented in SolFinder 1.0 to identify eco-efficient aircraft trajectories: application study in AirTraf 3.0

Federica Castino[1], Feijia Yin[1], Volker Grewe[1,2], Hiroshi Yamashita[2], Sigrun Matthes[2], Simone Dietmüller[2], Sabine Baumann[2], Manuel Soler[3], Abolfazl Simorgh[3], Maximilian Mendiguchia Meuser[4], Florian Linke[4], and Benjamin Lührs[4]

[1]Faculty of Aerospace Engineering, Delft University of Technology, Delft, The Netherlands
[2]Institut für Physik der Atmosphäre, Deutsches Zentrum für Luft- und Raumfahrt, Oberpfaffenhofen, Germany
[3]Department of Aerospace Engineering, Universidad Carlos III de Madrid, Madrid, Spain
[4]Lufttransportsysteme, Deutsches Zentrum für Luft- und Raumfahrt, Hamburg, Germany

**Correspondence:** Federica Castino (f.castino@tudelft.nl)

**Abstract.**

The optimization of aircraft trajectories involves balancing operating costs and climate impact, which are often conflicting objectives. To achieve compromise optimal solutions, higher-level information such as preferences of decision-makers must be taken into account. This paper introduces the SolFinder 1.0 module, a decision-making tool designed to identify eco-efficient aircraft trajectories, which allow the reduction of the flights climate impact with limited cost penalties compared to cost-optimal solutions. SolFinder 1.0 offers flexible decision-making options that allow users to select trade-offs between different objective functions, including fuel use, flight time, $NO_x$ emissions, contrail distance, and climate impact. The module is included in the AirTraf 3.0 submodel, which optimizes trajectories under atmospheric conditions simulated by the ECHAM/MESSy Atmospheric Chemistry model. This paper focuses on the ability of the module to identify eco-efficient trajectories while solving a bi-objective optimization problem that minimizes climate impact and operating costs. SolFinder 1.0 enables users to explore trajectory properties at varying locations of the Pareto fronts without prior knowledge of the problem results and to identify solutions that limit the cost of reducing the climate impact of a single flight.

## 1 Introduction

Aviation is estimated to contribute 3-5% of total anthropogenic global warming (Lee et al., 2021). Aircraft emissions affect the radiative budget of the Earth through several effects, which are usually distinguished between carbon dioxide ($CO_2$) and non-$CO_2$ effects (Lee et al., 2010). Non-$CO_2$ effects account for about 2/3 of the aviation net effective radiative forcing (Lee et al., 2021) and include, among others: the radiative forcing from contrail cirrus (Schumann, 2005; Kärcher, 2018), and the perturbations in the atmospheric concentrations of ozone ($O_3$) and methane ($CH_4$) caused by nitrogen oxide ($NO_x$) emissions (Stevenson et al., 2004; Köhler et al., 2008). The temperature perturbation resulting from $CO_2$ emissions is only dependent on the amount of emitted $CO_2$, due to the long atmospheric lifetime of $CO_2$. To reduce $CO_2$ emissions, several solutions are currently under development, exploring, for example, the use of new propulsion technologies or alternative aviation fuels

(Staples et al., 2018; Yin and Rao, 2020). Contrarily, non-$CO_2$ effects occur over short timescales, which typically range from hours (e.g., contrails) to months or years (e.g., $NO_x$-induced changes on $O_3$ and $CH_4$). As a consequence, the temperature perturbation caused by an aircraft unit emission is highly dependent on the time and location of the emission (Köhler et al., 2013; Frömming et al., 2021). Many studies investigated the possibility of using this time and space dependency to reduce the climate effect of a flight, for example, by optimizing its trajectory to minimize the induced temperature increase (Stevenson and Derwent, 2009; Sridhar et al., 2011; Grewe et al., 2017; Matthes et al., 2021). Towards the implementation of this type of operational strategies, some recent studies highlighted the main challenges and opportunities related to the avoidance of climate sensitive regions by aircraft trajectories optimization (Simorgh et al., 2022; Molloy et al., 2022). Most importantly, the current level of scientific understanding of the non-$CO_2$ effects of aviation is lower than the one of $CO_2$ effects, as demonstrated by the uncertainty ranges of the radiative forcing estimates reported by Lee et al. (2021). Moreover, the identification of climate sensitive regions (e.g., ice-supersaturated regions, supporting persistent contrails) relies on the availability of accurate weather forecast. Depending on the stability of the forecast, trajectories can be optimized via tactical adjustments during the flight or, preferably, in advance (e.g., one day before departure) to limit the associated penalties in operating costs, e.g., minimizing fuel use and workload of flight crew and air traffic controller (Molloy et al., 2022).

Currently, air traffic optimization focuses on minimizing economic penalties, e.g., identifying aircraft trajectories that lead to minimal operating cost. Minimizing the operating cost and the climate impact of a single flight are expected to be conflicting objectives of aircraft trajectory optimization (Yamashita et al., 2021). This implies that, in most cases, it is not possible to identify a solution that simultaneously minimizes both objectives. Therefore, optimizing an aircraft trajectory with respect to its economic cost and climate impact, a set of Pareto-optimal solutions can be identified. To select a single trajectory among this set of optimal solutions, a wide range of decision-making strategies can be employed (Kou et al., 2012; Pasman et al., 2022). In this paper, we present a new decision-making tool, the SolFinder 1.0 module, which was developed with the aim of identifying aircraft trajectories leading to a significant reduction of the flight climate impact, while limiting the increase in economic costs: we define these options as *eco-efficient* aircraft trajectories. This tool satisfies the following requirements:

- it is applicable to any set of Pareto-optimal solutions resulting from the optimization of a single aircraft trajectory;

- it is suitable for identifying compromise solutions between any number of objective functions;

- in particular, when applied for the bi-objective optimization of operating cost and climate impact, it is capable of identifying eco-efficient solutions.

To satisfy these requirements, the following options have been selected and are available in the first version of SolFinder: (1) a strategy relying on the VIKOR method (abbreviation from its Serbian name: *Vlse Kriterijumska Optimizacija Kompromisno Resenje*, presented by Opricovic and Tzeng, 2004) to identify eco-efficient solutions; (2) a strategy selecting the Pareto-optimal solution closest to a target percentage change in one of the objectives, such as the economic costs; (3) a decision-making method which combines the previous two options, applying the VIKOR method while limiting the change in one of the objectives. The SolFinder module has been coupled to the ECHAM/MESSy Atmospheric Chemistry (EMAC, Jöckel et al., 2010) submodel

AirTraf (Yamashita et al., 2020), as part of the AirTraf extension for the efficient resolution of multi-objective optimization problems. This modelling chain enables users to select Pareto-optimal solutions matching specific preferences of decision-makers, e.g., eco-efficient aircraft trajectories, and to explore their dependency on the atmospheric natural variability.

In Sect. 2, we describe the modelling chain, and we present the decision-making strategies included in SolFinder 1.0 (Sect. 2.3). In Sect. 3, we illustrate an example application of the selected decision-making strategies, using the Pareto-optimal solutions that are identified by AirTraf when a European air traffic sample of 100 night-time flights is optimized with respect to economic cost and climate impact. In Sect. 4, we compare our results to those obtained in previous studies, and we discuss uncertainties affecting our results. Our key messages are summarized in Sect. 5.

## 2   Methods

We conduct our simulations using the ECHAM/MESSy Atmospheric Chemistry (EMAC) model (Jöckel et al., 2010). This is a numerical climate model system that includes sub-models describing tropospheric and middle atmosphere processes and their interaction with oceans, land and human influences (Jöckel et al., 2010). This system relies on the second version of the Modular Earth Submodel System (MESSy2) to connect multi-institutional computer codes, while the core atmospheric model is the 5th generation European Centre Hamburg general circulation model (ECHAM5, Roeckner et al., 2006). Fig. 1 illustrates the relation between the EMAC model and the three submodels that have a major relevance in our experiments: CONTRAIL (Frömming et al., 2014), ACCF (Yin et al., 2023), and AirTraf (Yamashita et al., 2020).

The EMAC model provides the atmospheric conditions at a specific time and location (e.g., wind, temperature, potential vorticity, relative humidity) to determine the fuel consumption, emission indexes, and climate effects of aircraft emissions. The CONTRAIL submodel computes the potential contrail coverage, i.e., the fraction of the model grid-box where persistent contrails can exist (Burkhardt et al., 2008). The ACCF submodel employs the algorithmic Climate Change Functions (aCCFs) in order to deliver the estimated spatially and temporally resolved climate effect of aviation emissions to AirTraf; lastly, the AirTraf submodel identifies the optimal aircraft trajectories with respect to the routing strategy selected by the user. The optimization process is performed in two steps: (1) a genetic algorithm - Adaptive Range Multi-Objective Genetic Algorithms (ARMOGA, Sasaki and Obayashi, 2005) - is employed to solve a single- or multi-objective optimization problem; (2) if a multi-objective optimization is solved, it is possible that more than one optimal solution is found, thus a decision-making module (SolFinder) intervenes to select a single recommended trajectory, based on the preferences of the user.

### 2.1   Base model configuration

For the present study, we applied EMAC (ECHAM5 version 5.3.02, MESSy version 2.55.0) in the T42L31ECMWF-resolution. This resolution has a spherical truncation of T42 (corresponding to a quadratic Gaussian grid of approximately 2.8 by 2.8 degrees in latitude and longitude), and includes 31 vertical hybrid pressure levels up to 10 hPa (i.e., to an altitude of approximately 30 km). We describe the model output obtained by simulating the atmospheric conditions occurring from 1 January 2018 to 31 January 2018, employing a temporal resolution of 12 minutes. To obtain weather conditions aligned with those observed in

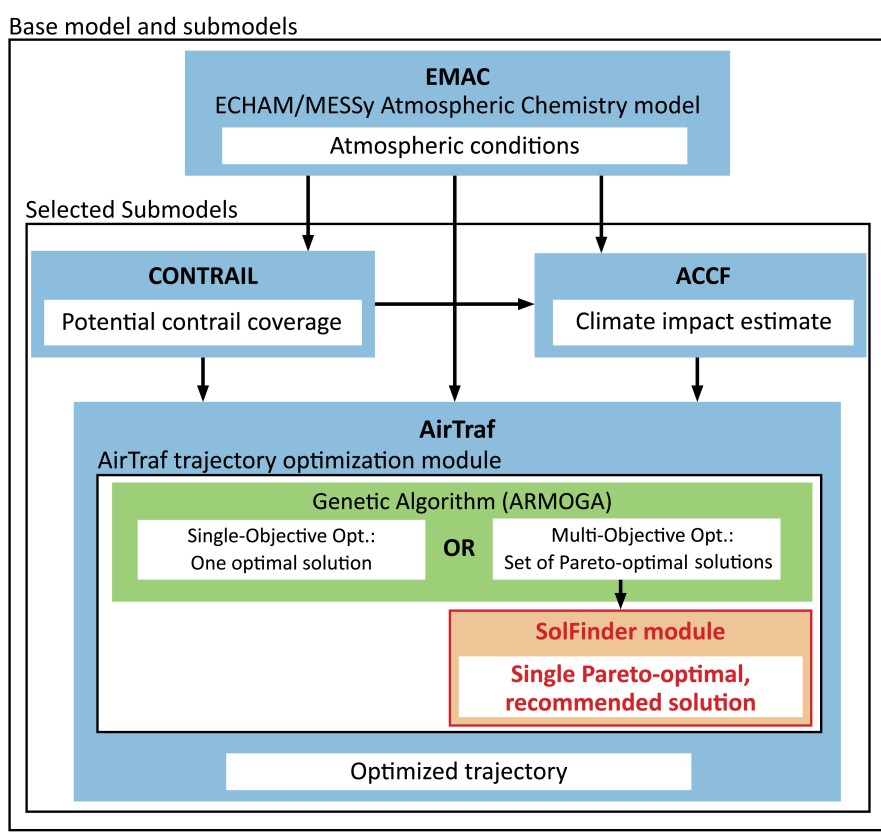

**Figure 1.** Overview of the relations between EMAC and the three submodels CONTRAIL, ACCF, and AirTraf. The present study focuses on the description of the decision-making module SolFinder 1.0, which is highlighted in red in the diagram.

January 2018, the simulations are conducted nudging by Newtonian relaxation the prognostics variables divergence, vorticity, temperature and the (logarithm of the) surface pressure down to the surface towards the respective ECMWF ERA-Interim reanalysis data (Dee et al., 2011; Jöckel et al., 2016).

## 2.2 AirTraf submodel

The air traffic simulator AirTraf is responsible for the optimization of the aircraft trajectories, according to the routing strategy prescribed by the user. The submodel requires as input information (1) the atmospheric conditions at the time and location of the flight, provided by the EMAC model, and (2) the air traffic sample, including the location of the airports of departure and arrival, the departure time of each flight, and characteristics of the aircraft and engine type to be simulated (Yamashita et al., 2016). Once this information is collected, the genetic algorithm (ARMOGA, Sasaki and Obayashi, 2005) intervenes to identify an optimal trajectory. The number of design variables of the optimization problem is fixed to 11, since the model describes each aircraft trajectory as a B-spline curve defined by three control points on the horizontal domain (three pairs of coordinates) and

five on the vertical cross-section (as illustrated in Yamashita et al., 2016, Fig.6). The domains of the horizontal control points are centered on the great circle connecting the airports of departure and arrival, while the vertical control points are bounded

by the flight levels at 29000 ft (FL290) and at 41000 ft (FL410), corresponding to altitudes of about 8.8-12.5 km. To calculate the flight properties along a candidate trajectory, the path is divided in flight segments by $n_{\text{wp}} = 101$ waypoints. In particular, to calculate the fuel used at each flight segment, AirTraf uses the aircraft performance model of Eurocontrol's Base of Aircraft Data (BADA Revision 3.9, Eurocontrol, 2011) and the Deutsches Zentrum für Luft- und Raumfahrt (DLR) fuel flow method (Yamashita et al., 2016).

The version AirTraf 2.0 presented by Yamashita et al. (2020) allowed the user to solve single-objective optimization problems, minimizing one of several available objective functions, including fuel use, flight time, $NO_x$ emissions, contrail distance, operating costs, and climate impact. The submodel is being expanded to use the same optimization method for the efficient resolution of multi-objective optimization problems. As a result, it is possible to simultaneously optimize two or more objective functions, without combining such functions into a single objective. This is particularly convenient when we aim to identify

eco-efficient aircraft trajectories under a large number of weather conditions. In fact, a conversion factor would be necessary to combine climate impact and operating costs into a single objective functions. However, this factor (e.g., as defined by Simorgh et al. (2023), Eq.7) is affected by a large variability over different flights and days, thus it is difficult to find a general value of the factor which can be applicable to optimize large air traffic samples over long-term simulations. Due to this AirTraf expansion for the resolution of multi-objective optimization problems, a decision-making module became necessary to ensure

that a single optimal solution is identified, and its properties are evaluated and stored by the model. This task can be performed by the SolFinder 1.0 module presented in this paper. Therefore, SolFinder has been coupled to AirTraf to select one optimal solution from the full Pareto set, reducing the computational effort of the model.

In the development of the decision-making strategies implemented in SolFinder 1.0, the underlying goal has been to find eco-efficient aircraft trajectories, compromising between the optimization of climate impact and operating costs. Hence, we

include here the definitions of these objective functions within AirTraf. The economic costs of the flights are represented by the Simple Operating Costs (SOC), defined as in Eq. (1):

$$\text{SOC} = c_t \sum_{i=1}^{n_{wp}-1} \text{TIME}_i + c_f \sum_{i=1}^{n_{wp}-1} \text{FUEL}_i \tag{1}$$

where $\text{TIME}_i$ and $\text{FUEL}_i$ represent the flight time and fuel used at the $i^{th}$ flight segment, respectively, while $c_t = 0.75 \text{ \$}/s$ and $c_f = 0.51 \text{ \$}/kg$ are the unit time and unit fuel costs (Burris, 2015; Yamashita et al., 2020). The climate impact of each aircraft

trajectory is measured in terms of Average Temperature Response over 20 years (ATR20), as provided by the ACCF submodel (van Manen and Grewe, 2019; Yin et al., 2023). The total climate impact $\text{ATR20}_{\text{tot}}$ of each aircraft trajectory is determined summing the contribution from the main climate effects:

$$\text{ATR20}_{\text{tot}} = \sum_{i=1}^{n_{wp}-1} \left[ \text{ATR20}_{\text{CO}_2,i} + \text{ATR20}_{\text{H}_2\text{O},i} + \text{ATR20}_{\text{NO}_x\text{-O}_3,i} + \text{ATR20}_{\text{NO}_x\text{-CH}_4,i} + \text{ATR20}_{\text{contrails},i} \right] \tag{2}$$

where each addend represents, from left to right, the ATR20 at the $i^{th}$ flight segment from (1) carbon dioxide ($CO_2$), (2) water vapour ($H_2O$), (3) ozone ($O_3$) from emission of $NO_x$, (4) methane ($CH_4$) from emission of $NO_x$, and (5) contrails. The term $ATR20_{NO_x\text{-}CH_4,i}$ includes the changes in primary mode ozone induced by the reduced $CH_4$ atmospheric concentration, while it neglects the feedback from stratospheric water vapour (Yin et al., 2023). The present study uses ATR20 as climate metric, assuming a business-as-usual Future emission scenario (F-ATR20). Alternative climate metrics can also be used, e.g, considering a time horizon of 100 years (ATR100). A detailed description of the climate metric conversion is presented in Dietmüller et al. (2023).

## 2.3 SolFinder module

In this section, we describe the decision-making strategies implemented in SolFinder 1.0. Our aim is to solve a multi-objective optimization problem minimizing a set of $N$ objective functions $f_n$, with $n = 1, 2, ..., N$. If two or more objective functions are conflicting, a set of Pareto-optimal solutions, $P$, is identified. The values $f_{n,j}$ of the objective functions are assigned to each Pareto-optimal solution $p_j \in P$ , with $j = 1, 2, ..., J$. Subsequently, a decision-making strategy intervenes, to select one solution $p_{\text{rec}} \in P$ which is the *recommended* solution according to the decision-maker criteria. We include the following strategies towards the resolution of our problem: (1) option selecting a solution based on its weighted distance from an ideal (usually, not feasible) solution (VIKOR method, Sect. A) for the identification of eco-efficient trajectories (Sect. 2.3.1); (2) option leading to a target percentage change in one of the objective functions, with respect to its minimum value (Sect. 2.3.2); (3) option combining the previous two strategies, limiting the change in one of the objective functions while applying the VIKOR method (Sect. 2.3.3); (4) selection of one of the extremes of the Pareto-optimal set (Sect. 2.3.4).

A large variety of multi-criteria decision-making methods is currently available to select one solution among a set of optimal options (Wang and Rangaiah, 2017; Sałabun et al., 2020). In a preliminary phase of our research, we considered different options among the most popular techniques, including GRA (Grey Relational Analysis, Wang and Rangaiah, 2017), TOPSIS (Technique for Order Preference by Similarity to Ideal Solution, Chen and Hwang, 1992), and VIKOR (abbreviation from its Serbian name: *Vlse Kriterijumska Optimizacija Kompromisno Resenje*, Sect. A). These options have been implemented in a python library[1], in order to apply them on test cases, and compare their effectiveness in identifying eco-efficient aircraft trajectories. We identified the VIKOR method (Opricovic, 1998; Opricovic and Tzeng, 2004) as a suitable candidate to translate our definition of eco-efficient solutions into a decision-making algorithm. This is due to its peculiarity of recommending more than one solution, if certain criteria are not met by a single solution (Sect. A). This allows flexible identification of a region of the Pareto-front (e.g., the section of the Pareto-front leading to a small change in the economic costs), within which the user is able to choose their preferred solution (e.g., the solution with the largest climate impact reduction). Moreover, no *a priori* knowledge on the optimization problem expected results is needed. To apply the VIKOR method, only the following information is provided as input, translating the user preferences in mathematical terms:

---

[1]See Supplement material and (Castino, 2023).

– the relative importance of $N$ objective functions, represented by the weights $w_n \geq 0$, such that $\sum_n w_n = 1$, with $n = 1, 2, ..., N$.

– the relative importance of group utility (preference towards achieving the greatest benefit) and individual regret (preference towards avoiding large penalties), represented by the parameter $\gamma$, with $0 < \gamma < 1$. If $\gamma > 0.5$, the *majority rule* principle is applied; contrarily, $\gamma < 0.5$ implies the application of the *veto* principle; lastly, $\gamma \approx 0.5$ represents a *voting by consensus* strategy (Opricovic and Tzeng, 2004).

### 2.3.1    Strategy using VIKOR method to identify eco-efficient trajectories

The SolFinder 1.0 module identifies eco-efficient trajectories using a decision-making option based on the VIKOR method. This method, introduced by Opricovic (1998) and Opricovic and Tzeng (2004), makes use of the overall distance (group utility) and the maximum distance (individual regret) of a Pareto-optimal solution from the minimal values of the optimization objec-
170 tives to rank the Pareto-optimal solutions. For more details on the version of the VIKOR method implemented in SolFinder, we refer to Sect. A. Fig. 2 illustrates how this decision-making strategy is applied using as an example the bi-objective optimization of a flight with respect to its SOC and ATR20. Firstly, the decision-making module collects all the Pareto-optimal solutions (represented by the grey crosses in Fig. 2a). Subsequently, the VIKOR method is applied according to the prescribed values of the parameter $\gamma$ and the relative weights $w_n$ (Fig. 2b). This leads to a recommended subset of optimal solutions (rep-
175 resented by the blue circles in Fig. 2c). If the VIKOR method identifies more than one recommended solution, i.e., the solutions $p_v$ $(v = 1, 2, ..., M)$ are equally recommended, the model selects the one with the minimum value of the objective function assigned to the lowest weight $w_n$. In Fig. 2d, the objective with the lowest relative weight is ATR20 ($w_{\text{ATR20}} = 0.3, w_{\text{SOC}} = 0.7$), thus the model selects the point among the recommended solutions (indicated by the blue circles in Fig. 2d) with the lowest ATR20 (red triangle in Fig. 2d). This last step is thus formulated to translate in mathematical terms our definition of eco-
180 efficient aircraft trajectories, i.e., a compromise solution between cost-optimal and climate-optimal solutions, such that the largest possible climate impact reduction is achieved, while keeping the operating costs nearly unchanged with respect to the cost-optimal solution. Using the VIKOR method, a subset of Pareto-optimal solutions is identified, according to the relative importance of the two optimization objectives. Therefore, if the highest weight is assigned to the objective function representing operating costs, the VIKOR method equally recommends a subset of Pareto-optimal solutions close to – or, possibly, including
185 - the cost-optimal extreme point of the Pareto-front. Among this subset of equally recommended solutions, we choose the point leading to the largest climate impact reduction, i.e., the minimum value of the objective function assigned to the lowest weight. Therefore, the objective with the highest weight plays a dominant role in the selection of the subset of equally recommended solutions (VIKOR method), while the objective with the lowest weight becomes dominant in the selection of a single solution among this subset.

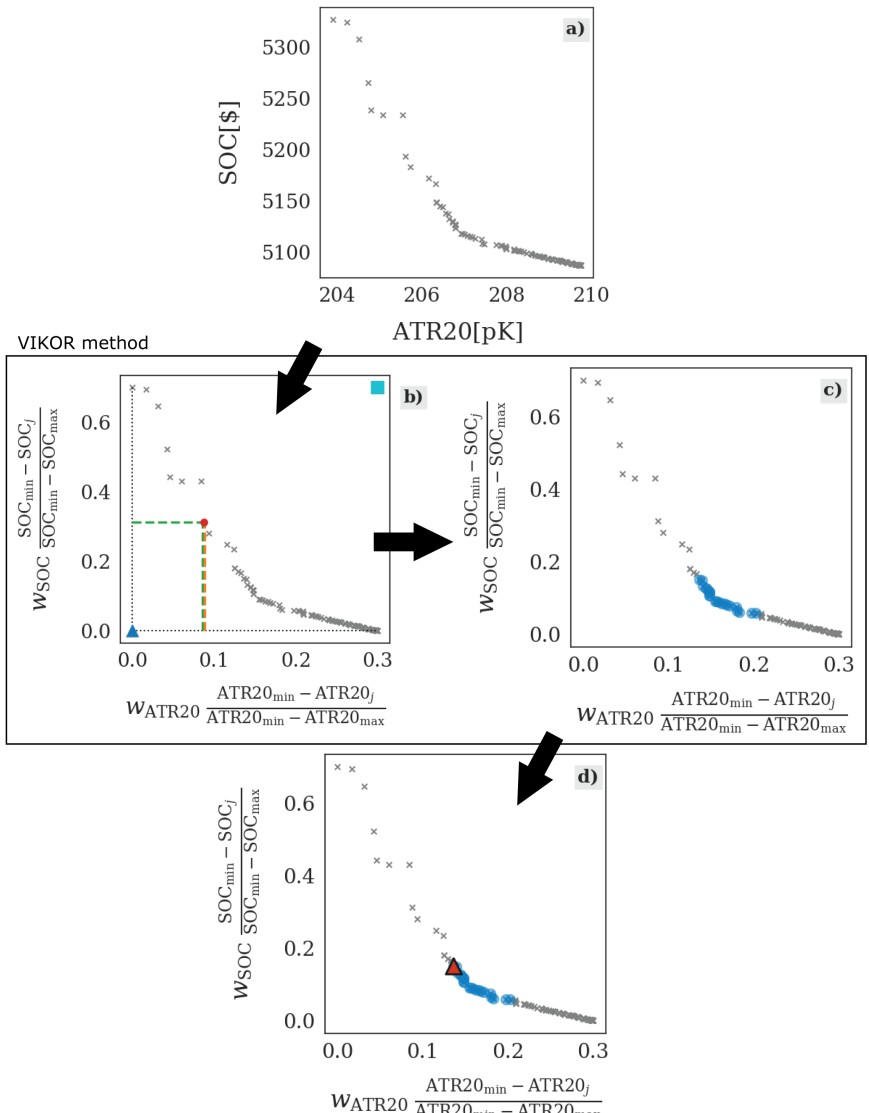

**Figure 2.** Illustration of the steps performed by the eco-efficient decision-making strategy relying of VIKOR. The aircraft trajectories are optimized to minimize SOC and ATR20, resulting in a set of Pareto-optimal solutions (grey crosses). We set $w_{SOC} = 0.7$, $w_{ATR20} = 0.3$, $\gamma = 0.5$. Panel a) shows the Pareto-optimal solutions (grey crosses) collected before applying the decision-making strategy. Panel b) illustrates the application of the VIKOR method (Sect. A), thus the axes are scaled as in Fig. A1. This step results in the identification of the subset of recommended solutions, represented by the blue circles in panel c). Panel d) shows the selected solution (red triangle) among the subset of recommended solutions (blue circles).

The resulting strategy can be configured to follow the steps listed here:

1. A bi-objective optimization problem is solved to simultaneously minimize the total climate impact (ATR20$_\text{tot}$ as defined in Eq. (2)) and operating costs (SOC, Eq. (1)). This step results in the identification of $J$ Pareto-optimal solutions (Fig. 2a).

2. The VIKOR method is applied, following the steps described in Sect. A (Fig. 2b).

3. A set of equally recommended solutions are selected in the sections in the Pareto front closest to the cost-optimal solution, by assigning a relatively high weight to the operating costs, i.e., $w_\text{SOC} > 0.5$ (with $\gamma = 0.5$). Depending on the shape of the Pareto front, this set of solutions extends towards the best ideal solution, $i_\text{best}$, allowing for higher climate impact reductions with respect to the cost-optimal solution, while avoiding cost penalties that are not compensated by a climate impact reduction (Fig. 2c).

4. Among this set of equally recommended solutions, the solution leading to the largest climate impact reduction with respect to the cost-optimal solution is selected, since $w_\text{ATR} < w_\text{SOC}$. We define this point as the *eco-efficient* solution among the set of Pareto-optimal options (Fig. 2d).

**Sensitivity of VIKOR parameterization**

To understand the effectiveness the VIKOR method with various configurations of $\gamma$ and $w$, we take as example a set of Pareto-optimal solutions, resulting from the bi-objective optimization of an aircraft trajectory with respect to the SOC and ATR20 of the flight. Within this set, a subset of solutions is *recommended* by the VIKOR method with different values of $\gamma$ and $w = [w_\text{SOC}, w_\text{ATR20}]$. The results of this analysis are shown in Fig. 3. This figure illustrates the impact of varying the weights $w$ by comparing different rows. It is possible to see how, increasing the value of $w_\text{SOC}$ from 0.2 (Fig.s 3a, 3b, 3c) to 0.8 (Fig.s 3j, 3k, 3l), the set of solutions recommended by VIKOR moves closer to the cost-optimal extreme of the Pareto front (0% change in SOC and ATR20). It is less intuitive to understand the impact of changing the parameter $\gamma$, which represents the relative importance of group utility. As explained in Sect. 2.3, and according to the formulas included in Sect. A, with $\gamma = 0.5$ the same relative importance is assigned to avoiding large penalties in one of the objectives, and to achieving the greatest overall benefit. In the results presented in Sect. 3.2, we always set the default value $\gamma = 0.5$. A value of $\gamma < 0.5$ leads to the application of the veto principle, i.e., if one of the objectives is heavily penalized by selecting a certain Pareto-optimal solution, then it will have a low likelihood to be recommended. Therefore, setting $\gamma = 0.25$ (as in Fig.s 3a, 3d, 3g, 3j) leads to the exclusion of elements located in the external sections of the Pareto-front, because of their distance to the opposite extreme of the Pareto front. On the other hand, when the veto principle is not applied ($\gamma \geq 0.5$), the set of recommended solutions (blue circles) can include the solution minimising the objective with the highest relative weight. For example, when $w_\text{SOC} = 0.8$ and $\gamma \geq 0.5$, the solution with minimum SOC is included in the set of Pareto-optimal solutions (Fig. 3k, 3l).

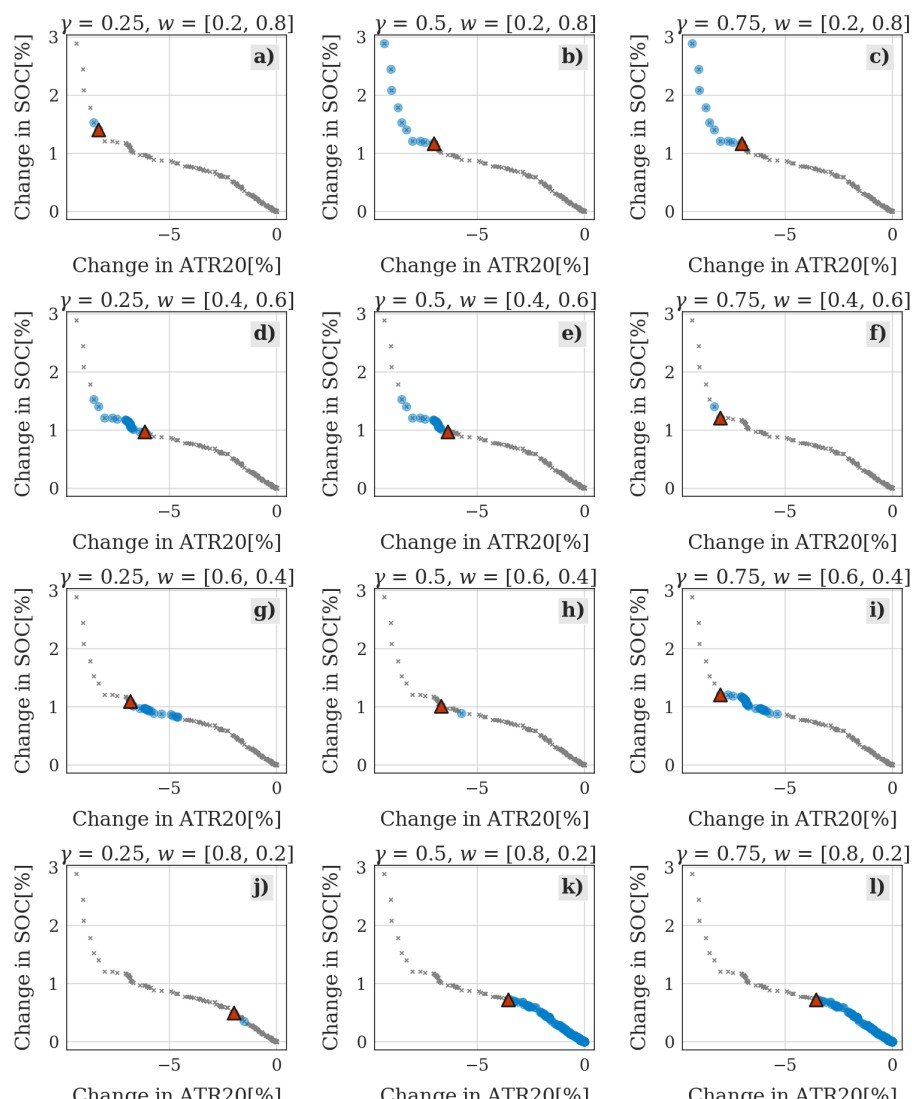

**Figure 3.** Variability of the selected solution (red triangle) using the eco-efficient decision-making method. The grey crosses represent the Pareto-optimal solutions, while the blue circles indicate the subset of solutions recommended by the VIKOR method. The axes show percentage changes in the objective functions, relative to the solution minimizing SOC. In this example, the Pareto front consists of 308 solutions.

This observation is the base for the definition of *eco-efficient* solution given at the end of Sect. 2.3.1, i.e., the solution selected with a high relative weight of SOC ($w_{SOC}$ larger than 0.5) and $\gamma = 0.5$. Following this definition, the selected solution shown in Fig. 3k (represented by a red triangle) is an eco-efficient solution. In fact, this solution leads to the maximum $ATR20_{tot}$ reduction among the subset of the solutions recommended by VIKOR (blue circles), while being equally recommended as the cost-optimal solution by the VIKOR method. In other words, the selected solution in Fig. 3k follows the definition of eco-

efficient solutions given in the Introduction, i.e., it allows for a significant reduction in the flight climate impact, while avoiding significant increases in operating costs. However, we note that determining the values of $\gamma$ and $w$ remains an arbitrary choice, which reflects the specific preferences of the decision-maker. Further elements to consider when setting these parameters are discussed later in Sect. 3.

### 2.3.2 Target percentage change in one of the objective functions

In some scenarios, the decision-maker wishes to limit the penalty of one of the objective functions, e.g., to avoid unsustainable increases in the operating costs. Therefore, in the second decision-making strategy, we propose to select the solution $p_{\mathrm{rec}}$ leading to a percentage change in one of the objective functions, $f_p$. On the other hand, to obtain the expected outcome, this option requires certain information on the optimization results to be known before solving the problem, such as the typical shape of the Pareto fronts, and reference values at their extreme points. The threshold for the allowable change is specified as a target value $x_t$. This is translated in the following process:

$$f_{p,\mathrm{ref}} = \min_j f_{p,j}, \qquad j = 1, 2, ..., J \tag{3}$$

$$x_j = 100 * \frac{f_{p,j} - f_{p,\mathrm{ref}}}{f_{p,\mathrm{ref}}} \tag{4}$$

$$\Delta x_j = |x_j - x_t| \tag{5}$$

Therefore, the selected Pareto-optimal solution $p_{\mathrm{rec}} \in P$ is the solution corresponding to the relative change $x_{\mathrm{rec}}$, such that $\Delta x_{\mathrm{rec}} = \min_j \Delta x_j$.

Fig. 4 illustrates which solution is selected within a Pareto-optimal set resulting from the resolution of a tri-objective optimization problem, simultaneously minimizing flight time, fuel use, and ATR20$_{\mathrm{tot}}$. In this example, $x_t = +0.5\%$ is set as the target percentage change in flight time.

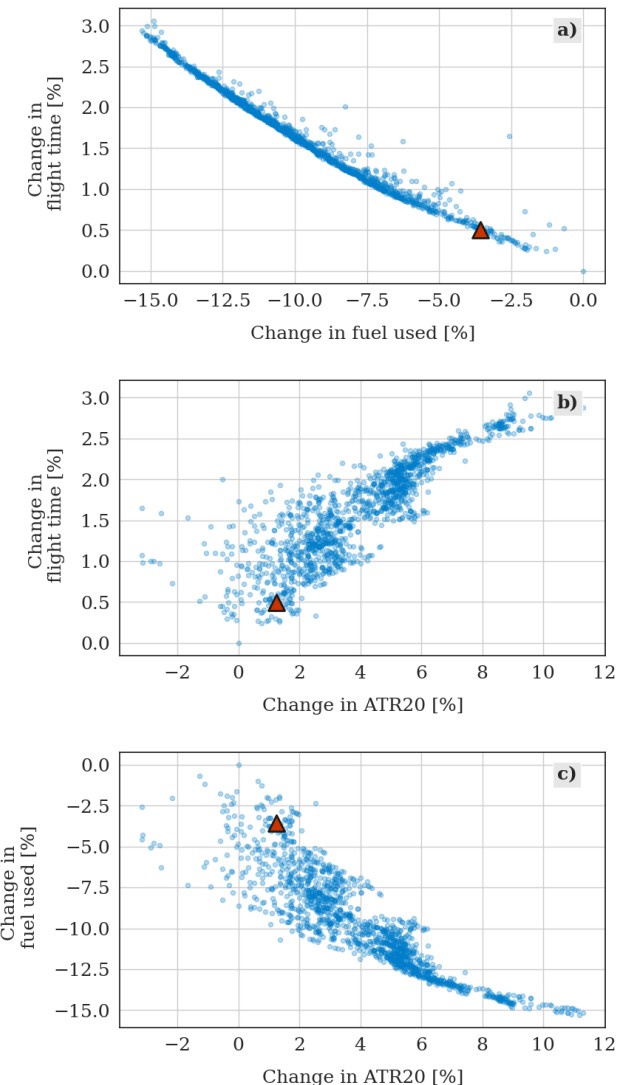

**Figure 4.** Example of selecting the solution among the Pareto-surface matching a target increase in 0.5% in flight time (indicated by red triangles). The blue circles indicate the Pareto-optimal solutions, which result from a tri-objective optimization problem minimizing flight time, fuel use, and ATR20$_{tot}$.

### 2.3.3 Hybrid option: VIKOR method with target percentage change in one of the objectives

To combine the advantages of the two decision-making strategies presented, a hybrid option is considered, limiting the variability in one of the objective functions while applying the VIKOR method. When this strategy is selected, the decision-maker provides: (1) the configuration of the VIKOR method, setting the parameters $\gamma$ and $w$, and (2) a target percentage increase $x_t$ in the objective $f_p$. Subsequently, the following decision-making process is followed:

1. Apply the VIKOR method and select the recommended solution minimizing the objective function having the lowest weight $w_n$, as described in Sect. A.

2. Calculate $x_p$, representing the relative change in $f_p$ of the recommended solution with respect to the minimum value of $f_p$.

3. If $x_p > x_t$, replace the recommended solution identified in step 1. with the the Pareto-optimal solution leading to a target percentage change $x_t$ in the objective function $f_p$ (in other words, apply the strategy described in Sect. 2.3.2).

This strategy addresses the fact that, with the VIKOR method, no limit in the percentage increase of the objective is set, thus a fraction of the solutions can be affected by changes much larger than the average, as is shown in a later section of this paper (Sect. 3.2, Fig. 7). With this hybrid option, the VIKOR method can be employed to identify eco-efficient trajectories, while introducing a constraint on the operating costs, to prevent increases in the operating costs of some flights that the decision-maker does not accept.

### 2.3.4  Selecting one extreme of Pareto-optimal set

Lastly, an additional option is considered to select a solution minimizing one of the objective functions $f_n$, which we indicate as $f_{\min}$. This simple decision-making process selects the optimal-solution corresponding to the result of a single-objective optimization minimizing $f_{\min}$. Nevertheless, it can be useful to implement this method to identify which values are used as reference during the resolution of the multi-objective optimization problem, and to verify the performance of the model.

## 3  Application of decision-making method to analyse trajectories' variability along Pareto-front

We now present an example study, in which different settings of the decision-making strategies are compared. This application exemplifies how the decision-making strategies can be employed, and what to consider to determine the settings that best translate the decision-maker preference. In this example, we focus on the suitable settings to identify eco-efficient aircraft trajectories. Nevertheless, SolFinder can also be used to comply with alternative decision-making preferences, by changing the settings of AirTraf and SolFinder.

### 3.1  Simulations set-up

As previously stated, we intend to identify eco-efficient aircraft trajectories, i.e., trajectories reducing the climate impact at limited changes in the operating costs. Therefore, we solve a bi-objective optimization problem, aiming to simultaneously minimize SOC and ATR20, as defined in Sect. 2.2. We conduct one-month simulations, from 1 January 2018 to 31 January 2018. On each simulation day, 100 night-time flights departing at 00:00 UTC are optimized. Fig. 5 shows the locations of the airports of origin and destination, which were selected considering the Available Seat Kilometres (ASK) for the European Civil Aviation Conference (ECAC) area in 2018 (Meuser et al., 2022). The same criterion was used to select the A320-214

(CFM56-5B4) as the aircraft/engine type to be simulated. The climate impact of each aircraft trajectory is estimated using the aCCF 1.0A (Matthes et al., 2023). Table 1 summarizes the model configuration.

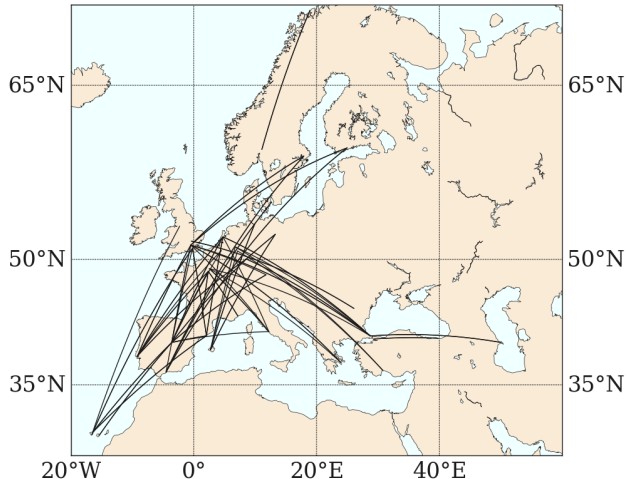

**Figure 5.** Location of the 100 flights included in the air traffic sample. Each curve represents the great circle connecting an origin/destination pair. Note that most origin/destination pairs are connected by two flights, i.e., one for each direction, thus the number of curves is lower than 100. The list of ICAO airport codes is included in Table B1.

To compare the effects of using different decision-making strategies, we perform two sets of experiments, whose characteristics are summarized in Table 2, and explained below:

1. In the first set of experiments, the VIKOR method is employed as we described in Sect. 2.3.1, fixing $\gamma = 0.5$ and varying the relative weight $w_{\mathrm{SOC}}$ between 0.2 and 0.9 (Table 2). Therefore, we obtain a total of six selected trade-off solutions. The density of $w_{\mathrm{SOC}}$ values increases for higher values of $w_{\mathrm{SOC}}$, as these values lead to the selections of solutions in the section of the Pareto-front of major interest when searching for eco-efficient trajectories, that is, the section closer to the cost-optimal solution. Additionally, we select the two solutions located at the Pareto-front extremes, i.e., the cost-optimal and climate-optimal solutions, with the routine mentioned in Sect. 2.3.4. No preliminary knowledge about the expected problem results is needed to conduct these simulations. As a result of this first set of experiments, for each optimized flight we obtain information on eight solutions that determine the shape and extension of the Pareto-fronts relative to individual flights, and the relation between penalties and benefit aggregated over the whole air traffic sample.

2. In the second set of experiments, we explore the effects of selecting a solution leading to a target change in SOC (Sect. 2.3.2). The target values are chosen using the results form the first set of experiments, which obtained average SOC increases up to about +3.0% (see Table 3 and Fig. 6). In light of these results, we vary $x_t$ from 0.5% to 3.0% (Table 2), selecting four solutions per flight. Moreover, we set $w_{\mathrm{SOC}} = 0.7$ to exemplify the effects of using the option of using VIKOR while setting a target SOC change (Sect. 2.3.3) of +1.0%.

**Table 1.** Main settings of ECHAM5, ACCF, and AirTraf.

| | **ECHAM5** |
|---|---|
| Horizontal resolution | T42 (2.81° × 2.81°) |
| Vertical resolution | L31ECMWF (31 vertical pressure levels up to 10 hPa ∼ 30 km ) |
| Time step | 12 min |
| Duration | 1-31 Jan. 2018 (each day, 1 month) |
| | **ACCF** |
| Version aCCFs | V1.0A |
| Climate metric | F-ATR20 |
| Forcing efficacy | Included |
| | **AirTraf** |
| Trajectory waypoints | 101 |
| Aircraft / Engine | A320-214 (CFM56-5B4) |
| Air Traffic Sample | Top 100 routes by ASK for the ECAC area in 2018 |
| Departure time | 00:00 |
| Optimization strategy | Multi-objective optimization of: (1) SOC, (2) ATR20$_{tot}$ |

## 3.2 Results

As explained in Sect. 2.3, one of the main advantages of the VIKOR method is that no knowledge of the expected results is needed before solving our problem. For this reason, the first set of experiments applies the VIKOR method with a range of values of the parameter $w_{SOC}$. This enables us to perform a preliminary examination of the characteristics of the relation between benefit (in our application, ATR20$_{tot}$ reduction) and penalty (increase in SOC). Relative to our problem, this relation is shown in Fig. 6, illustrating the change in SOC that is required to achieve a certain reduction in ATR20. The grey curves in Fig. 6a are obtained aggregating the 100 Pareto-optimal solutions selected using the VIKOR method on each simulation day, varying the relative weight $w_{SOC}$ of the objective function SOC. The coordinates of the points connected by the black

**Table 2.** Overview of the two sets of experiments performed to exemplify the use of the different decision-making methods.

| Set | SolFinder strategy | Num. selected sol. | Parameter | Description | Value(s) |
|---|---|---|---|---|---|
| 1 | **VIKOR method for eco-eff.** | 6 | $\gamma$ | Group utility weight | 0.5 |
| | | | $w_{SOC}$ | Rel. weight of SOC | 0.2, 0.4, 0.6, 0.7, 0.8, 0.9 |
| | | | $w_{ATR}$ | Rel. weight of ATR20$_{tot}$ | $1 - w_{SOC}$ |
| | **Pareto-front extremes** | 2 | $f_{min}$ | Minimized objective | ATR20$_{tot}$ or SOC |
| 2 | **Target change in $f_p$** | 4 | $f_p$ | - | SOC |
| | | | $x_t$ | Target change [%] | 0.5, 1.0, 2.0, 3.0 |
| | **Hybrid VIKOR-target option** | 1 | $\gamma$ | Group utility weight (VIKOR) | 0.5 |
| | | | $w_{SOC}$ | Rel. weight of SOC (VIKOR) | 0.7 |
| | | | $w_{ATR}$ | Rel. weight of ATR20$_{tot}$ (VIKOR) | $1 - w_{SOC}$ |
| | | | $f_p$ | objective with limited changes | SOC |
| | | | $x_t$ | Limit and target change [%] | 1.0 |

line (representing the average relation over the 31 simulated days) can be found in Table 3. Setting a low value of $w_{SOC}$ (e.g., $w_{SOC} = 0.2$), we obtain a reduction in ATR20$_{tot}$ of about 13%, which is almost as large as the maximum potential reduction achieved selecting a climate-optimal routing strategy. Moving along the curve towards the cost-optimal strategy, the magnitude of the ATR20$_{tot}$ reduction decreases. This occurs with a simultaneous decrease of the cost per Kelvin reduction in climate impact, measured by the climate-cost coefficient $k$ [\$/K] (Table 3). This coefficient is defined as (Matthes et al., 2017):

$$k = \frac{\text{SOC} - \text{SOC}_{\text{cost-opt}}}{\text{ATR20}_{\text{cost-opt}} - \text{ATR20}} \tag{6}$$

where ATR20$_{\text{cost-opt}}$ and SOC$_{\text{cost-opt}}$ are relative to the cost-optimal solution, while ATR20 and SOC are relative to the considered decision-making strategy. From this first set of experiments, we understand that: (1) the average $\Delta$ATR20$_{tot}$ ranges from -3.5 to -14.4%, with an increase in SOC of 0.1-3.0%, respectively; (2) about half of the maximum feasible climate change reduction can be achieved with only +0.5% in SOC.

As previously mentioned, the red and green points highlighted in Fig. 6a result from summing all the solutions selected for the 100 flights, and averaging over the month of simulation. However, each optimized trajectory is characterised by a different change in ATR20$_{tot}$ and SOC with respect to its corresponding cost-optimal solution. We note here that this cost-optimal solution (shown in red in Fig. 6a), which we take as reference to calculate the relative changes in ATR20$_{tot}$ and SOC, is specific to each route and each simulation day, thus it varies between different flights. The full variability of the relationship between relative changes in ATR20$_{tot}$ and SOC is illustrated in Fig. 6b. In this panel, one can see that specific solutions can show large deviations from the average values, as some solutions reach an absolute $\Delta$ATR20$_{tot}$ larger than 60% at penalties

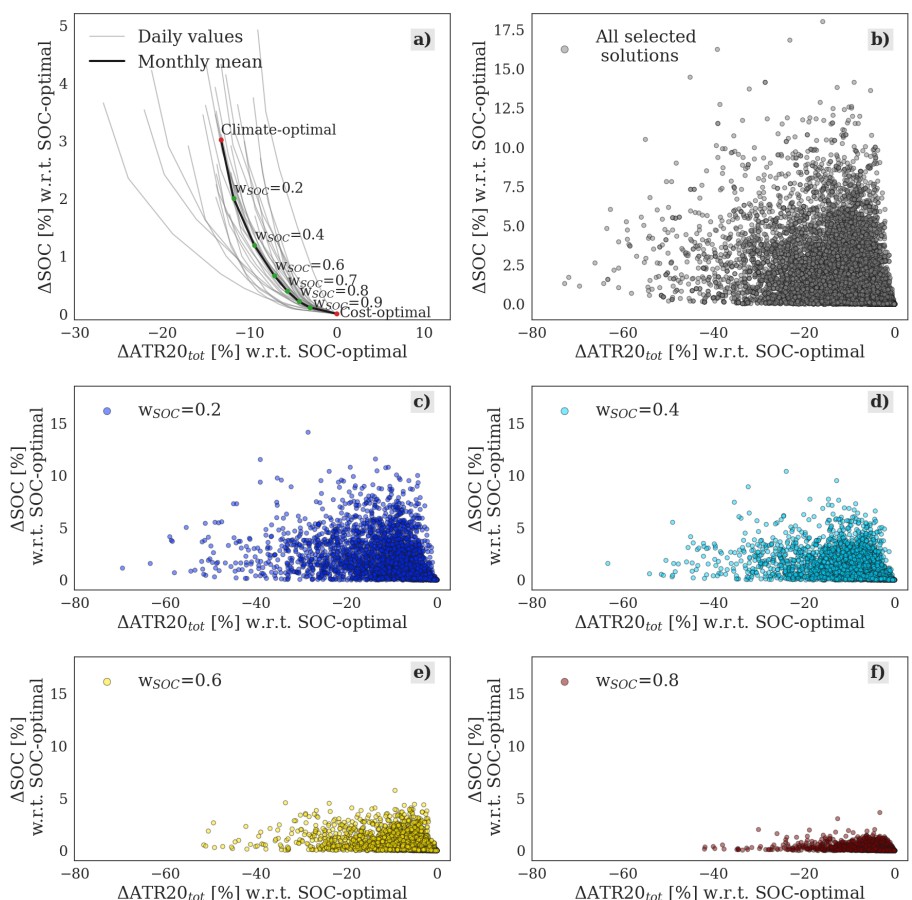

**Figure 6.** Relation between the relative changes in climate impact, $\Delta$ATR20$_\text{tot}$ [%], and in simple operating cost, $\Delta$SOC [%], with respect to the cost-optimal solution. *Panel a):* Values obtained summing over the 100 routes optimized per day. The black line illustrates the average values over the 31 days included in the simulations, connecting the points selected varying the VIKOR weight $w_\text{SOC}$ from 0.2 to 0.9 (green dots). The extremes of the Pareto fronts (climate- and cost-optimal solutions, red dots) are included. The gray lines represent the Pareto fronts obtained on each simulation day. *Panels b)-f):* Scatter graphs of $\Delta$ATR20$_\text{tot}$ [%] against $\Delta$SOC [%], representing all the individual selected solutions (6b), and subsets of solutions obtained varying the weight of simple operating costs, $w_\text{SOC}$ (6c-6f).

lower than 2.5% in terms of $\Delta$SOC. Moreover, Fig. 6c-f show that the subsets of Pareto-optimal solutions obtained setting different $w_\text{SOC}$ behave as expected: moving from lower to higher values of $w_\text{SOC}$ leads to subsets that are confined below lower increases in SOC, while they still stretch towards high ATR20$_\text{tot}$ reductions (e.g., see the points selected with $w_\text{SOC}$=0.8 in Fig. 6f). The distributions of $\Delta$ATR20$_\text{tot}$ and $\Delta$SOC values across the optimized flights are represented in Fig. 7, which compares
the results obtained varying the parameter $w_\text{SOC}$ applying the VIKOR method, and the climate-optimal scenario. Fig. 7b shows the percentage of flights characterized by a certain $\Delta$SOC. One can see that, employing VIKOR, the mode of each curve is

close to $\Delta SOC \sim 0\%$, while the mean and maximum $\Delta SOC$ values increase when the weight $w_{SOC}$ decreases. Looking at the $\Delta ATR20_{tot}$ distributions in Fig. 7a, we can observe larger differences also in the modal values for varying $w_{SOC}$.

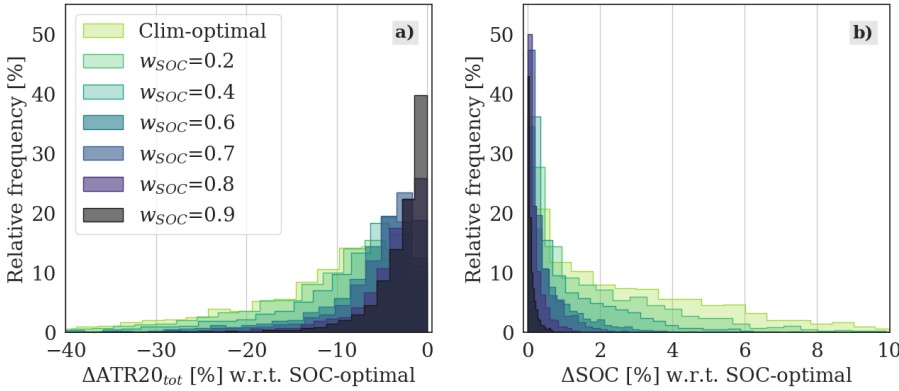

**Figure 7.** Relative frequencies [%] of different values of $\Delta ATR20_{tot}$ [%] (Fig. 7a), and of $\Delta SOC$ [%] (Fig. 7b) with respect to the cost-optimal solution. The histograms compare the distributions of the values obtained with different decision-making strategies, considering the 100 flights optimized on each simulation day (31*100 values per histogram). The solutions are selected by identifying the minimum climate impact (climate-optimal solutions), or employing the VIKOR method with varying $w_{SOC}$ (first set of experiments).

The results of the first set of experiments can then be used to conduct the second set, fixing a target or a limit increase in
SOC as described in Sect. 3.1. The result of this second set of experiments are shown in Table 4. To illustrate the difference of employing different decision-making methods, it is useful to compare Fig. 7 with Fig. 8. The most evident difference emerges comparing Fig. 7b with Fig. 8b. As intended, when we set a target increase in SOC (e.g., +1.0%), most of the selected flights are affected by an increase in SOC equal or similar to the target, thus the curves are centered on this $\Delta SOC$ value. However, we can also deduce that some Pareto-fronts do not extend to the target $\Delta SOC$, since values of $\Delta SOC$ which are lower than
the target are assigned to a fraction of the flights. This results in average $\Delta SOC$ values that are lower than the targets, as we can see in Table 4. For example, a total increase of 0.8% in SOC is obtained, when the target is set to 1.0%. Moreover, Fig. 8b shows what distribution of $\Delta SOC$ the user can expect employing the hybrid method described in Sect. 2.3.3. The histogram relative to the hybrid option (yellow, hatch-filled) corresponds to the one obtained with VIKOR (with $w_{SOC}$=0.7), with an additional peak at the determined target SOC increase (1.0%), replacing the larger SOC increases observed in Fig. 7b. The
results illustrated in Figs. 7b and 8b confirm that the strategies available in SolFinder 1.0 are correctly implemented, and lead to the expected selection of Pareto-optimal solutions. Moreover, the possibility of relying on the VIKOR method to identify eco-efficient trajectories is confirmed by Fig. 9. These curves represent the distributions of the climate-cost coefficient $k$ [\$/K], obtained using VIKOR (blue curves) or the target SOC change (red curves). It is possible to observe that lower values of $k$ are obtained using VIKOR, in particular when a relatively high weight is assigned to SOC.
Employing different decision-making strategies, we obtain trajectories which are characterized by different properties. How these properties vary is shown in Fig. 10, which compares the mean flight altitudes and flown distances obtained when different

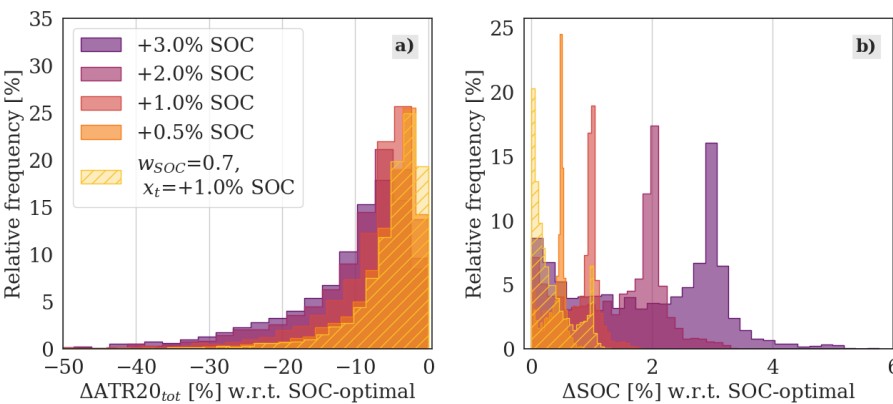

**Figure 8.** Relative frequencies [%] of different values of $\Delta$ATR20$_{\mathrm{tot}}$ [%] (Fig. 8a), and of $\Delta$SOC [%] (Fig. 8b) with respect to the cost-optimal solution. The histograms compare the distributions of the values obtained with different decision-making strategies, considering the 100 flights optimized on each simulation day (31*100 values per histogram). The solutions are selected by targeting different SOC changes, or applying the hybrid method with $w_{\mathrm{SOC}}$=0.7 and $x_t$ = 1.0% (second set of experiments).

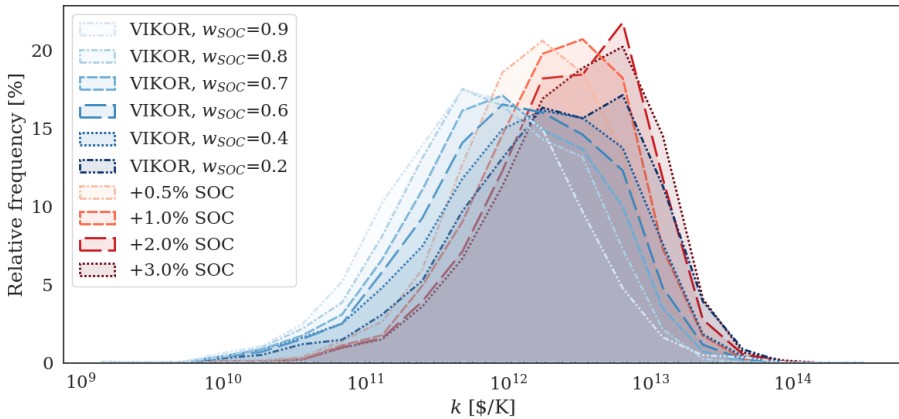

**Figure 9.** Relative frequencies [%] of different values of the climate-cost coefficient $k$ [$/K], comparing the SolFinder solution-picking strategies using VIKOR (blue curves) or the target SOC change (red curves). The curves approximate the histogram outlines (connecting the bars centers) to highlight the shapes of the distributions and facilitate their comparison. Each curve includes the values obtained with different decision-making strategies, considering the 100 flights optimized on each simulation day (31*100 values per histogram).

decision-making strategies are selected. Firstly, we can notice that cost-optimal flights are characterized by the highest mean flight altitudes and the shortest trajectories among the solutions considered, due to the presence of fuel consumption in the optimization objective. On the opposite situation, the lowest altitude and the longest distances are obtained for climate-optimized flights. This confirms previous studies, which commented that (1) aircraft emissions have lower climate impact at lower altitudes, e.g., due to shorter residence time of emitted species (Castino et al., 2021; Matthes et al., 2021; Frömming et al., 2012),

and (2) lateral deviations may be necessary to avoid climate sensitive regions (Matthes et al., 2020). Moreover, the variability of the flight properties across the set of simulated flights is larger for climate-optimized flights than for cost-optimal ones. For example, comparing the different variability in flight altitudes, we see the impact of the lowest aerodynamic drag, which allows minimal fuel use, is systematically found at higher altitudes. Contrarily, the altitude leading to the minimal ATR20 is highly variable, due to the high temporal and spatial variability of the atmospheric conditions determining the net flight ATR20. Trade-off solutions between these two extreme scenarios show intermediate properties, with median values and interval bars which gradually evolve moving from one extreme of the curve in Fig. 6 to the other. We can also see that employing the VIKOR method rather than fixing a target increase in costs can lead to different tendencies in the average characteristics of the selected trajectories. For example, comparing the VIKOR method with $w = 0.4$ (causing $\Delta SOC \sim +1.2\%$, see Table 3) and setting a target +1.0% change in SOC, we can see that the latter strategy leads to a less frequent selection of lower flight altitudes, while flying longer trajectories, than the former case. Therefore, the user is recommended to identify their preferred decision-making strategy by also considering these secondary aspects, and not exclusively the resulting distributions in objective functions values.

Lastly, we analyse the contribution to the total change in ATR20 of each effect of aviation emissions that we considered in our optimization process: $CO_2$, $H_2O$, contrails, $NO_x$ via perturbation of ozone, and $NO_x$ via methane depletion. The relative importance of these effects under different decision-making strategies is illustrated in Fig. 11. Firstly, we can notice that the ATR20 from $CO_2$ emissions increases moving from cost-optimal to climate-optimal and compromise solutions. This increase is expected, as the climate impact of $CO_2$ is independent of the background atmospheric conditions at time and location of emission; thus it is simply proportional to the amount of fuel used, which is minimized by cost-optimal flights. On the other hand, the increase in the ATR20 from $CO_2$ is largely compensated by the reduction in the ATR20 from contrails and, secondarily, from the impact of $NO_x$ on ozone. Moreover, Fig. 11 shows that reduction in climate impact of contrails becomes increasingly dominant over the reduction of the other effects, when the relative weight $w_{SOC}$ of SOC increases. This aspect should be considered when selecting the preferred settings of the decision-making strategy. For example, if the decision-maker is interested in reducing both the climate impact of contrails and $NO_x$-ozone via trajectory optimization, this goal can be achieved allowing larger changes in SOC than those needed to only reduce the ATR20 from contrails effects. The dominant contributions of contrails and, secondarily, of the $NO_x$ climate impact to the ATR20 reduction is confirmed by the results obtained optimizing aircraft trajectories with different tools (Lührs et al., 2021; Simorgh et al., 2022). However, large uncertainties affect the estimates of non-$CO_2$ effects of aviation (see Sect. 1 and 4), which can heavily affect the relative importance of the different effects of aviation emissions. Therefore, it is relevant to update the modelling chain presented here when enhancements in the scientific understanding and in our ability to model non-$CO_2$ effects are available, for example, via updated aCCFs versions.

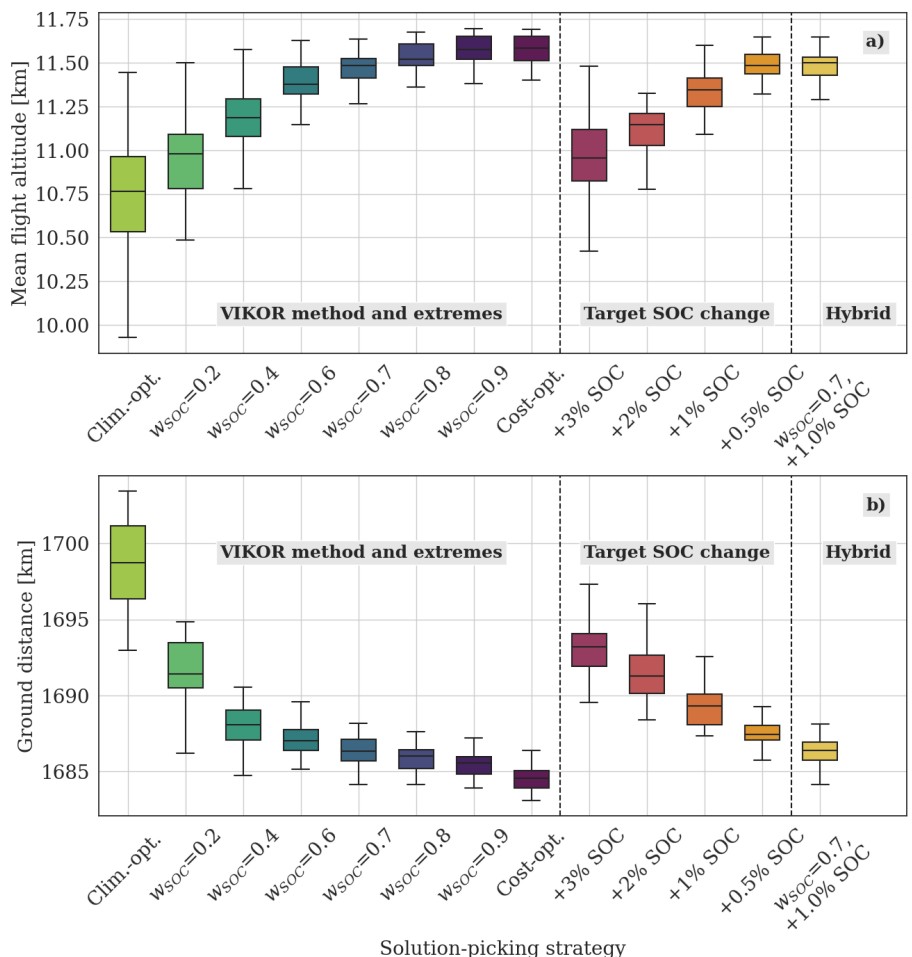

**Figure 10.** Variability of mean flight altitudes [km] (Fig. 10a) and ground distance [km] (Fig. 10b) when employing different decision-making strategies. The flight altitudes are calculated averaging over each trajectory, and over the 100 flights optimized on each day. The ground distances represent the total length of each trajectory, averaged over the 100 flights optimized on each simulation day. The boxes extend between 1st and 3rd quartiles over time, and include segments marking the median values. The whiskers indicate the the distribution of the remaining values (excluding eventual outliers).

## 4 Discussion

In this paper, we illustrated the decision-making strategies implemented in the SolFinder 1.0 module, and how they can be
385   used to identify eco-efficient trajectories. The climate-optimization of aircraft trajectories has been increasingly researched in the last decade, as efforts to reduce the climate impact of aviation lead to the investigation of operational mitigation strategies. For example, Grewe et al. (2017) optimized a set of transatlantic flights during eight representative weather patterns in winter and summer, employing the air traffic simulator SAAM and the Climate Change Functions (CCFs). This study found that

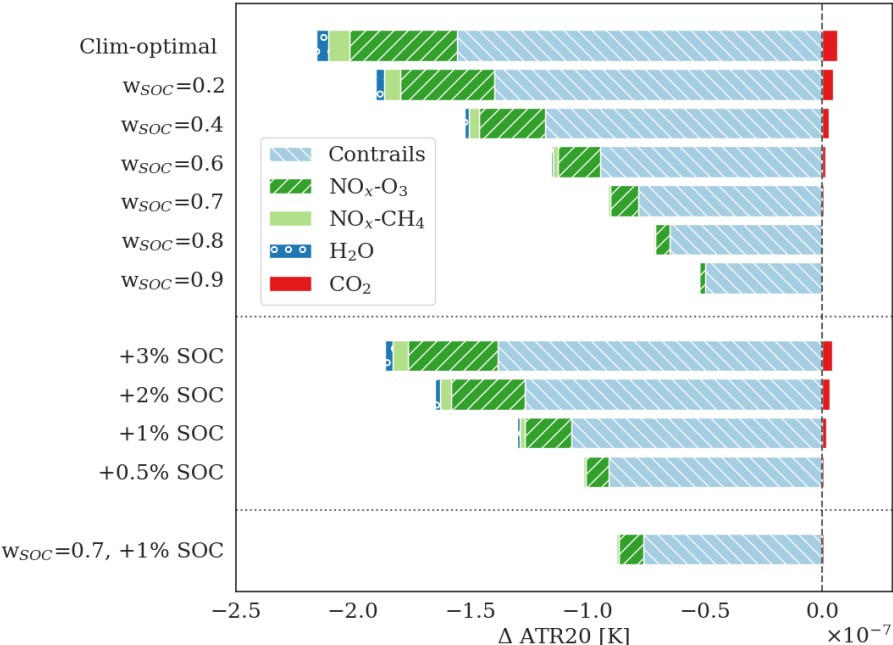

**Figure 11.** Contribution of different climate effects of aviation to the absolute $\Delta$ATR20 [K] when employing different decision-making strategies. The horizontal, dotted line distinguish between solutions selected using VIKOR or climate-optimal solutions (upper section), solutions selected targeting a SOC relative change (middle section), and solutions selected applying the hybrid method (lowest bar). The vertical, dashed line separates positive (warming) from negative (cooling) values of $\Delta$ATR20.

a 10% reduction in climate impact can be achieved with a 1.0% increase in operating costs. In our study, we considered a different region of the airspace (European flights), and we included a larger number of weather patterns (every day in the month of January 2018), but limited to the winter season and to night-time; nevertheless, we found the results in Grewe et al. (2017) to be aligned with those we presented in Sect. 3, since a $\Delta$SOC $\sim +1.2\%$ corresponds to a $\Delta$ATR20 $\sim -10.3\%$ (Table 3). The maximum feasible climate impact reduction is higher (about 20%) in Grewe et al. (2017), but this is achieved with an approximately double increase in costs. The discrepancy in the section of the Pareto-front closer to the climate-optimal solution may be due to the different temporal and spatial domains. The mitigation potential of optimized aircraft trajectories is expected to be affected by seasonal and daily variability (e.g., Reutter et al. (2020); Castino et al. (2021)), thus the methodology presented in this paper can be extended to cover multiple years, to investigate this aspect. Under specific weather conditions, cost-benefit relations were found to show higher eco-efficiency than were found aggregating the model output over all the routes, and all the simulated days. For example, Lührs et al. (2021) analysed a day characterised by a weather situation with strong contrail formation, and found that a 0.75% increase in fuel used halved the climate impact. This study suggests the possibility of identifying weather situations that allow for a higher eco-efficiency than others. The modelling chain presented in this paper enables us to optimize aircraft trajectories under a large number of different atmospheric conditions, within a

feasible computational time. Therefore, on-going research is using the newly developed version of AirTraf, coupled with the SolFinder module, to analyse under which weather conditions eco-efficient aircraft trajectories are most likely to be identified.

405 To this end, additional decision-making strategies are being investigated, to exploit the ability of VIKOR to identify lower values of the climate-cost coefficient $k$ than a strategy applying a target increase in SOC to all flights (Fig. 9). Two candidate additional strategies are illustrated in Fig. 12. With these options, only a fraction of the flights identified by VIKOR is climate-optimized, due to an additional condition. This is obtained setting a threshold value of the coefficient $k$, in order to climate-optimize only the top ranked half of the flights (Fig. 12a) or until a certain budget is spent (e.g, +0.5% of SOC in Fig. 12b). A

410 disadvantage of these decision-making strategies is that their settings configuration relies on information on the whole system of optimized flights, which is not available when we consider the Pareto front resulting from a single-flight optimization. Therefore, preliminary simulations have to be run to derive some parameter values, for example the specific threshold value of the climate-cost coefficient $k$ [\$/K]. Because of the large variability of $k$, these decision-making strategies are more reliant on the results obtained from previous simulations than the strategies included in SolFinder 1.0.

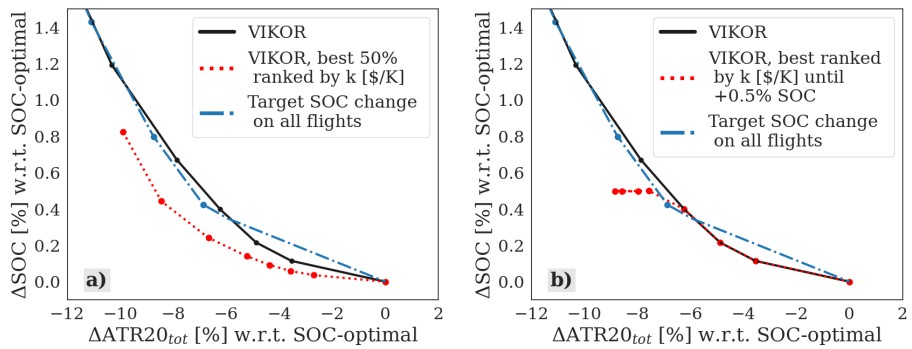

**Figure 12.** Relation between the percentage changes in climate impact, $\Delta$ATR20 [%], and in simple operating cost, $\Delta$SOC [%], with respect to the cost-optimal solution. The values are obtained aggregating the properties of all the optimized flights (31 days, 100 routes optimized per day). The black curve connects the points obtained with the SolFinder strategy relying on VIKOR, while the blue line and points refer to the SolFinder strategy targeting a fixed SOC change. The red lines refer to additional strategies considered for future versions of SolFinder: optimization of the 50% of the flights best ranked by $k$ (Fig. 12a), and optimization of the flights best ranked by $k$ until an increase in SOC of 0.5% is achieved.

415 The present work estimates the climate effect of aviation resulting from the emission of $CO_2$, $H_2O$, $NO_x$, and from the formation of contrail-cirrus. Estimating the radiative forcing caused by non-$CO_2$ effects is a complex process, leading to results that are affected by large uncertainties due to, for example, incomplete scientific understanding and modeling capabilities (Lee et al., 2021). As described in Sect. 3.1, we use the aCCFs version 1.0A to estimate the climate impact of aviation (Matthes et al., 2023), which are calibrated towards the results of a climate response model (AirClim, Dahlmann et al., 2016) to align the

420 relative importance of individual aCCFs. This is an update of the consistent set of aCCF 1.0 (Yin et al., 2023), calculating the climate impact of $CO_2$, $H_2O$, $NO_x$-ozone, $NO_x$-methane and contrails in terms of ATR20, assuming pulse emissions (P-ATR20). This prototype set of functions are the focus of on-going research, to address their sources of uncertainties. Moreover, we

employed factors to: (1) convert the aCCFs values to a different climate metric, F-ATR20, which assumes a business-as-usual future emission scenario; (2) include the efficacy of each climate impact effect (Dietmüller et al., 2023). These assumptions introduce additional sources of uncertainties (Dietmüller et al., 2023). Moreover, the majority of ice-supersaturated regions (supporting persistent contrail formation) have characteristic dimensions that are smaller than the horizontal resolution of our model. To take this factor into account, we employ a parameterization developed by Burkhardt et al. (2008) to estimate the fraction of model grid box which is supporting persistent contrails. The potential of reducing the flights climate impact by contrail avoidance could be reduced by factors not included in this study. For example, here we use weather conditions simulated by an atmospheric model (EMAC) rather than weather forecast. In real life applications, this mitigation strategy relies on the accuracy and stability of weather forecast, as discussed in Sect. 1. Lastly, we note that we use a simplified representation of the operating cost, to limit the computational time required for their evaluation within each optimization step. To this end, we assume a linear relationship between cost of time and flight time (see Eq. (1) and Yamashita et al., 2020). Therefore, we neglect additional costs caused by delays. Further research will explore the impact of optimizing not only the location of the flight trajectory (see Sect. 2.2), but also the airspeed, improving fuel efficiency compared to this study, which assumed a constant Mach number for all solutions. Other studies found that speed changes can be important for reducing fuel flow, $NO_x$ emissions and, ultimately, $NO_x$ climate effects (Simorgh et al., 2023). This improvement of AirTraf is expected to impact the SolFinder results, by reducing the penalties in terms of operating costs and $CO_2$ emissions for a certain gain in terms of reduction of ATR20. On the other hand, including the airspace structure and capacity, which are neglected in this study, can reduce the estimated climate impact mitigation potential of this operational strategy.

## 5 Conclusions

In this study, we described the decision-making strategies implemented in the SolFinder 1.0 module. The SolFinder 1.0 module has been coupled to the AirTraf 3.0 submodel, as part of its development to efficiently solve multi-objective optimization problems. We showed here how the selected decision-making strategies can be used to identify solutions matching specific preferences (e.g., eco-efficient aircraft trajectories). Moreover, using this modelling chain, it is possible to explore the results variability under a large number of consecutive days, due to the coupling between SolFinder and an atmospheric chemistry model (EMAC), via the EMAC submodel AirTraf. To demonstrate the usage of the tool, this paper showed results for the period of one winter month (1-31 January 2018). We solved a bi-objective optimization problem minimizing the climate impact of the aircraft trajectory (F-ATR20$_{tot}$) and its simple operating costs (SOC), and we compared the solutions selected by different configurations of SolFinder 1.0. Comparing the strategies using VIKOR and a target change in SOC, we found that lower values of the climate-cost coefficient $k$ [\$/K] (i.e., a higher eco-efficiency) are obtained with the former option. The decision-making strategies included in SolFinder 1.0 are applied on sets of Pareto-optimal solutions relative to a single aircraft trajectory. In the next SolFinder versions, we plan to take into account the mitigation potential variability across all flights. As a result, only the best performing fraction of the flights is optimized with respect to their climate impact, and the cost of the operational mitigation strategy is lowered. On-going research is using the modelling chain presented in this paper

to identify those weather situations allowing for the largest reductions in the temperature response from aviation emissions via the optimization of aircraft trajectories.

*Code and data availability.* The Modular Earth Submodel System (MESSy) is continuously further developed and applied by a consortium of institutions. The usage of MESSy and access to the source code is licenced to all affiliates of institutions which are members of the
MESSy Consortium. Institutions can become a member of the MESSy Consortium by signing the MESSy Memorandum of Understanding. More information can be found on the MESSy Consortium Website (http://www.messy-interface.org). The code presented here has been based on MESSy version 2.55.0 and will be available after the official release of AirTraf 3.0, a submodel of MESSy. An open access version of SolFinder (see Supplement material) is available from the 4TU.ResearchData repository (Castino, 2023) under the licence GNU Lesser General Public License v3.0, as are the scripts to produce the plots presented in this paper. The simulation output analysed in this paper is
archived in the 4TU.ResearchData repository (Castino et al., 2023).

## Appendix A: VIKOR method

In this Sect. A, we quote the steps characterizing the VIKOR method, as they were introduced and described in Opricovic (1998) and Opricovic and Tzeng (2004). Fig. A1 illustrates the main working principles of the VIKOR method. The x-axis represents the normalized distance from the minimum value of $f_1$, scaled by its relative weight $w_1$. As an example, this weight is
set to $w_1 = 0.2$, thus the x-axis ranges from 0 to 0.2. Similarly, the y-axis represents the normalized distance from the minimum value of $f_2$, weighted using $w_2 = 0.8$. The axes intersect at the reference point $i_\text{best}$, defined as the best *ideal* (i.e., usually not feasible) solution. Opposite to $i_\text{best}$, it is possible to identify $i_\text{worst}$, which assumes the worst values of $f_n$ ($f_{n,\text{worst}}$) found among the set of Pareto-optimal solutions. For example, when aiming at minimizing $f_n$, $i_\text{worst}$ corresponds to the maximum $f_n$ among the Pareto set. The grey points in Fig. A1 indicate the location of the Pareto-optimal solutions, relative to $i_\text{best}$. In fact, the
VIKOR method ranks the Pareto-optimal solutions using $i_\text{best}$ as a reference point. Hence, the first step consists of identifying such a reference point, by determining the best values of each of the objectives, $f_{n,\text{best}}$. Since we aim to minimize the objective functions, we use the following definitions of $f_{n,\text{best}}$ and $f_{n,\text{worst}}$:

$$f_{n,\text{best}} = \min_j f_{n,j}, \qquad f_{n,\text{worst}} = \max_j f_{n,j}, \qquad n = 1, 2, ..., N, \qquad j = 1, 2, ..., J \tag{A1}$$

In Fig. A1 we highlight the Pareto-optimal solution $p_j$. In the second step of the VIKOR method, two quantities are calculated
for each point in the Pareto set: $S(p_j)$, which measures the *group utility* of the solution $p_j$, and $R(p_j)$, which represents its *individual regret*. In other words, $S(p_j)$ measures the overall distance of $p_j$ from $i_\text{best}$, taking into account all the optimization objectives. On the other hand, $R(p_j)$ measures the largest distance of $p_j$ from $i_\text{best}$ considering each objective individually. These quantities are defined by Eqs. (4) and (5), respectively:

$$S(p_j) = \sum_{n=1}^{N} w_n \frac{f_{n,\text{best}} - f_{n,j}}{f_{n,\text{best}} - f_{n,\text{worst}}} \tag{A2}$$

$$R(p_j) = \max_n \left[ w_n \frac{f_{n,\text{best}} - f_{n,j}}{f_{n,\text{best}} - f_{n,\text{worst}}} \right] \tag{A3}$$

where $w_n$ is the relative weight of each objective $f_n$.

The geometric representation of $S(p_j)$ and $R(p_j)$ is illustrated in Fig. A1. It is possible to deduce from Eqs. (4) and (5) and from Fig. A1 that lower values of $S(p_j)$ and $R(p_j)$ are preferable. These measures of the distance of $p_j$ from $i_{\text{best}}$ are combined in the value $Q(p_j)$, which is used as main ranking parameter by the VIKOR method. The value of $Q(p_j)$ is calculated using

Eq. (6):

$$Q(p_j) = \gamma \frac{S_j - \min_j S_j}{\max_j S_j - \min_j S_j} + (1 - \gamma) \frac{R_j - \min_j R_j}{\max_j R_j - \min_j R_j} \tag{A4}$$

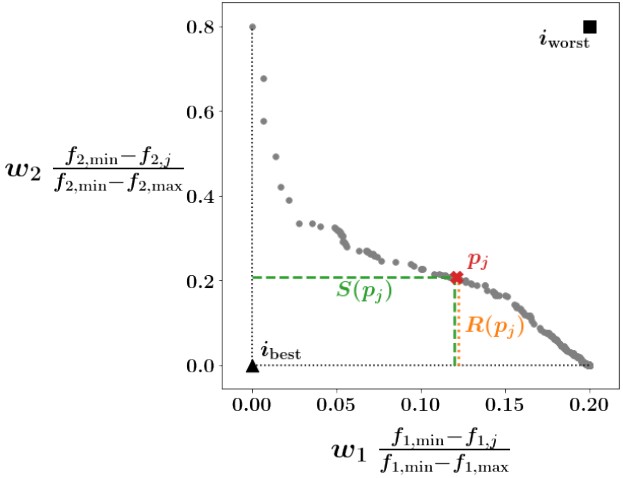

**Figure A1.** Illustration of the VIKOR method applied to a bi-objective optimization problem, minimizing $f_1$ and $f_2$. In this example, we set $w_1 = 0.2$ and $w_2 = 0.8$, thus the x (y) axis ranges from 0.0 to 0.2 (0.8). The grey dots represent Pareto-optimal solutions. The red cross indicates the Pareto-optimal solution $p_j$. The green dashed line represents $S(p_j)$, while the orange dotted segment represents $R(p_j)$. The reference points $i_{\text{best}}$ and $i_{\text{worst}}$ are indicated by the black triangle and black square, respectively.

The next step consists in creating three ranking lists of the Pareto-optimal solutions, sorting them by $S$, $R$, and $Q$. We define $p_i$ as the Pareto-optimal solution at the $i^{\text{th}}$ position in the list sorted by $Q$. Consequently, the first compromise solution to be recommended is $p_1$, which minimizes Q: $Q(p_1) = \min_j Q(p_j)$. The following conditions are then evaluated:

1. **acceptable advantage**: $Q(p_2) - Q(p_1) \geq \frac{1}{J-1}$

If this condition is not verified, a set of Pareto-optimal solutions $p_v$ $(v = 1, 2, ..., M)$ is recommended, where $M$ is the maximum value for which $Q(p_M) - Q(p_1) \leq \frac{1}{J-1}$ is true.

2. **acceptable stability**: $p_1$ is the best ranked solution not only by $Q$, but also by $S$ and $R$.

   If this condition is not satisfied, both $p_1$ and $p_2$ are recommended.

Therefore, the application of the VIKOR method results in the identification of either one optimal solution, $p_1$, or a subset of optimal solutions, $p_v$ $(v = 1, 2, ..., M)$, which are recommended to the decision-maker.

# Appendix B: Air traffic sample

| citypairs | ICAO airport code | | citypairs | ICAO airport code | |
|---|---|---|---|---|---|
| | departure | arrival | | departure | arrival |
| 0 | LTFM | EGLL | 50 | EDDL | LTFM |
| 1 | EGLL | LTFM | 51 | LTFM | EDDL |
| 2 | LEMD | GCLP | 52 | LFPO | LFBO |
| 3 | GCLP | LEMD | 53 | LEMD | EHAM |
| 4 | GCXO | LEMD | 54 | LFBO | LFPO |
| 5 | LEMD | GCXO | 55 | EHAM | LEMD |
| 6 | LFPG | LTFM | 56 | LFMN | LFPO |
| 7 | LTFM | LFPG | 57 | LFPO | LFMN |
| 8 | EGLL | LEMD | 58 | EGLL | LPPT |
| 9 | LFPO | LPPT | 59 | LPPT | EGLL |
| 10 | LPPT | LFPO | 60 | LEBL | LEMD |
| 11 | LEMD | EGLL | 61 | LFPG | LIRF |
| 12 | LEPA | EDDL | 62 | LIRF | LFPG |
| 13 | EDDL | LEPA | 63 | EHAM | LIRF |
| 14 | EHAM | LTFM | 64 | LIRF | EHAM |
| 15 | LTFM | EHAM | 65 | LEPA | EDDT |
| 16 | LGAV | EGLL | 66 | EDDT | LEPA |
| 17 | EGLL | LGAV | 67 | LEMD | LFPO |
| 18 | LEBL | EGKK | 68 | LFPO | LEMD |
| 19 | EGKK | LEBL | 69 | EDDM | EGLL |
| 20 | LEMD | LIRF | 70 | EGLL | EDDM |
| 21 | LIRF | LEMD | 71 | LEMD | LEBL |
| 22 | ESSA | EGLL | 72 | UBBB | LTFM |
| 23 | EGLL | ESSA | 73 | LTCG | LTFJ |
| 24 | EHAM | LEBL | 74 | LTFJ | LTCG |
| 25 | LEBL | EHAM | 75 | LFPG | LGAV |
| 26 | EDDF | LEMD | 76 | LGAV | LFPG |
| 27 | EGCC | GCTS | 77 | LEMD | EDDM |
| 28 | LEMD | EDDF | 78 | EDDM | LEMD |
| 29 | GCTS | EGCC | 79 | ESSA | LEMG |
| 30 | LPPT | EDDF | 80 | LEMG | ESSA |
| 31 | EDDF | LPPT | 81 | LEMD | EBBR |
| 32 | LIRF | EGLL | 82 | EBBR | LEMD |
| 33 | EGLL | LIRF | 83 | LROP | EGGW |
| 34 | LPPT | EHAM | 84 | EGGW | LROP |
| 35 | EHAM | LPPT | 85 | LPPT | EBBR |
| 36 | ENGM | ENTC | 86 | LTFM | UBBB |
| 37 | LTFM | EDDF | 87 | EBBR | LPPT |
| 38 | EDDF | LTFM | 88 | LEMG | EFHK |
| 39 | EGKK | LEMG | 89 | EFHK | LEMG |
| 40 | ENTC | ENGM | 90 | GCXO | LEBL |
| 41 | LEMG | EGKK | 91 | LEBL | GCXO |
| 42 | EGLL | EFHK | 92 | LTAI | EDDK |
| 43 | EFHK | EGLL | 93 | EDDK | LEPA |
| 44 | LTAI | EDDL | 94 | LEPA | EDDK |
| 45 | EDDL | LTAI | 95 | EDDH | LEPA |
| 46 | GCTS | EGKK | 96 | EDDT | EDDF |
| 47 | EGKK | GCTS | 97 | LEPA | EDDH |
| 48 | LEMG | EKCH | 98 | EDDF | EDDT |
| 49 | EKCH | LEMG | 99 | LPPR | LFPO |

**Table B1.** List of origin/destination airport pairs included in the air traffic sample, as illustrated in the map in Fig. 5.

*Author contributions.* FC, FY, VG, and HY developed the concepts presented in this paper. FC and HY implemented the decision-making module in the AirTraf submodel. FC performed the simulations and analysed the results presented in this paper. SM and SD provided the aCCFs formulas and factors employed in this study. BL, FL, and MMM provided the air traffic sample employed this study. All co-authors contributed to the discussion and the revision of the paper.

*Competing interests.* At least one of the (co-)authors is a member of the editorial board of Geoscientific Model Development. The peer-review process was guided by an independent editor, and the authors also have no other competing interests to declare.

*Acknowledgements.* The current study has been supported by FlyATM4E project, which has received funding from the SESAR Joint Under-taking (JU) under grant agreement No 891317. The JU receives support from the European Union's Horizon 2020 research and innovation programme and the SESAR JU members other than the Union. The computing resources to conduct simulations with the ECHAM/MESSy Atmospheric Chemistry (EMAC) model were provided by the TU Delft High Performance Cluster (HPC12).

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

**Table 3.** Properties of the trajectories selected within the 1st set of experiments. The table provides the total monthly percentage changes in climate impact, $\Delta \text{ATR20}_{\text{tot}}$ [%], in simple operating cost, $\Delta \text{SOC}$ [%], in fuel used, $\Delta \text{Fuel}$ [%], and in flight time, $\Delta \text{Time}$ [%], relative to the respective total values from the cost-optimal scenario; and values of the climate-cost coefficient $k$ [$/K]. The minimum and maximum daily values are included within brackets.

| SolFinder Strategy | $\Delta \text{ATR20}$ [%] | $\Delta \text{SOC}$ [%] | $\Delta \text{Fuel}$ [%] | $\Delta \text{Time}$ [%] | Coeff. $k$ [$/K] |
|---|---|---|---|---|---|
| Climate-optimal | -14.4(-29.2, -6.7) | +3.04(+1.32, +5.00) | +8.14(+2.41,+15.19) | +1.01(+0.59, +1.52) | 3.95e+12( 1.58e+12, 1.33e+13) |
| VIKOR $w_{\text{SOC}}$=0.2 | -12.8(-26.0, -5.9) | +2.00(+0.72, +3.52) | +5.72(+1.38,+11.18) | +0.51(+0.07, +0.98) | 2.94e+12( 1.07e+12, 1.05e+13) |
| VIKOR $w_{\text{SOC}}$=0.4 | -10.3(-21.1, -4.7) | +1.19(+0.42, +2.56) | +3.50(+0.70, +8.28) | +0.27(+0.01, +0.57) | 2.21e+12( 6.05e+11, 9.37e+12) |
| VIKOR $w_{\text{SOC}}$=0.6 | -7.9(-13.8, -4.0) | +0.67(+0.21, +1.52) | +1.86(+0.30, +5.07) | +0.20(-0.01, +0.34) | 1.66e+12( 4.65e+11, 8.10e+12) |
| VIKOR $w_{\text{SOC}}$=0.7 | -6.2(-11.0, -2.8) | +0.40(+0.14, +0.74) | +1.09(+0.19, +2.61) | +0.12(-0.04, +0.27) | 1.27e+12( 3.71e+11, 6.54e+12) |
| VIKOR $w_{\text{SOC}}$=0.8 | -4.9(-8.7, -1.3) | +0.22(+0.09, +0.34) | +0.55(+0.05, +1.30) | +0.08(-0.09, +0.22) | 8.81e+11( 2.37e+11, 4.31e+12) |
| VIKOR $w_{\text{SOC}}$=0.9 | -3.5(-6.3, -0.5) | +0.11(+0.06, +0.18) | +0.25(-0.00, +0.46) | +0.06(-0.05, +0.16) | 6.46e+11( 1.80e+11, 2.85e+12) |

**Table 4.** Properties of the trajectories selected within the 2$^{nd}$ set of experiments. The table provides the total monthly percentage changes in climate impact, $\Delta$ATR20 [%], in simple operating cost, $\Delta$SOC [%], in fuel used, $\Delta$Fuel [%], and in flight time, $\Delta$Time [%], relative to the respective total values of the cost-optimal scenario; and values of the climate-cost coefficient $k$ [\$/K]. The minimum and maximum daily values are included within brackets.

| SolFinder Strategy | $\Delta$ATR20 [%] | $\Delta$SOC [%] | $\Delta$Fuel [%] | $\Delta$Time [%] | Coeff. k [\$/K] |
|---|---|---|---|---|---|
| +3.0% SOC | -12.5( -25.2, -6.3) | +1.92( +1.00, +2.82) | +5.47( +1.79, +9.55) | +0.51(-0.05, +0.91) | 2.90e+12( 1.22e+12, 9.67e+12) |
| +2.0% SOC | -11.1( -21.5, -5.7) | +1.43( +0.82, +1.96) | +4.04( +1.34, +6.94) | +0.39( -0.11, +0.72) | 2.44e+12( 9.40e+11, 8.45e+12) |
| +1.0% SOC | -8.7( -14.9, -3.4) | +0.80( +0.59, +0.98) | +2.14( +0.82, +3.59) | +0.26( -0.13, +0.52) | 1.79e+12( 6.72e+11, 7.08e+12) |
| +0.5% SOC | -6.9( -12.8, -2.0) | +0.42( +0.36, +0.50) | +1.02( +0.39, +1.90) | +0.19( -0.06, +0.36) | 1.27e+12( 3.89e+11, 6.13e+12) |
| VIKOR $w$=0.7, +1.0% SOC | -6.0( -10.7, -2.4) | +0.35( +0.14, +0.61) | +0.94( +0.19, +2.21) | +0.11( -0.05, +0.24) | 1.18e+12( 3.52e+11, 6.38e+12) |