# Peer review of "Decision-making strategies implemented in SolFinder 1.0 to identify eco-efficient aircraft trajectories: application study in AirTraf 3.0"

_Geoscientific Model Development, 2023_

## Referee Comment (RC1)

The paper is relevant to the area of aircraft routing and addresses the choice between multiple optimal routes dependent on a range of objectives. As such this is not a major change to routing strategies, but rather a next step in an evolving process. This means that the novelty of the approach lies in applying an established algorithm to a new setting. The advance this affords allows the previously defined AirTraf model to be used in a different and more integrated way. The method described is mostly clear, but some of the assumptions supporting the use of AirTraf in contrail avoidance are not properly justified, given research results previously published on the nature of super saturated icy regions. This is further explored in the more detailed file that is attached below.

Results are somewhat limited by the use of a single month, which when looking at climatic conditions provides a narrower range of possible variable value combinations than is most useful. However, conclusions across this reduced timeframe given the other model assumptions are supported by the research that has been completed. Given the reliance of the results on a combination of different models and the limited explanation in this paper of the climate inputs and methods for fuel use calculation, it would be difficult to reproduce results from this work alone, but taken alongside previous research and given the limited access to models available, replication of some of the results could be possible.

The work is properly referenced and the need for the model is justified, with the paper title including all necessary detail. The abstract is concise and reflects the paper content, but could be better worded (see attached comments). The main structure of the paper is good, but there are occasions where order within sections would be improved by small changes. The language and grammar of the paper need minor corrections, which have been noted in the attached comments. Clear definitions of formulae, symbols and abbreviations are, however, all present in the paper.

Overall my recommendation is for the paper to be accepted subject to minor revisions. Where assumptions weaken the usefulness of the model, this stems more from the original AirTraf usage than the current decision-making tool, so whilst these choices do need more justification in this paper, they are not grounds for major revisions.

Comments on gmd-2023-88

The paper is recommended for publication with minor revisions. These are listed below. It is hoped that this extra level of scrutiny is of use to the authors in improving the manuscript before publication.
1) Abstract: lines 4-6 "This paper…solutions". This phrase does not read properly. Suggest: "…which allow the reduction of the flights'…"
2) Line 14: "to 3-5% of the total", the "to" and the "the" are not required.
3) Line 26: None of the references that have been included alluded to work on minimising just carbon dioxide emissions. Given the uncertainty of non-carbon dioxide effects, which is not alluded to until the Discussion Section, this should be mentioned here. There is also no justification for the use of strategic rather than tactical planning for avoidance of ice super saturated regions (ISSR) to prevent contrail formation, which often cannot be accurately forecast pre-flight (Reutter 2020).

4) Lines 35-42: These would read better if requirements and solutions were given together, rather than listing all requirements and then all solutions. A list structure would also make the arguments clearer, rather having numbered points lost in a passage of prose.

5) Line 45: "This modelling chain allows to select…" does not make sense as it stands. Allows who to select?

6) Line 68: Space needed after the comma.

7) Line 73-76: This description is very unclear. What do you mean by nudging? Please explain in more detail exactly which climate indices are used and how these are processed.

8) Section 2.1: Using reanalysis data does not take into account the difficulty in forecasting ISSR, which should be acknowledged. Seasonal patterns in ISSR formation are also not included when a single month is considered. In addition, Reutter et al (2020) show that ERA Interim reanalysis data misrepresents ice supersaturation at flight altitudes and that there is a significantly larger fraction of ISSR in measurements of water vapour. A resolution of 2.8 degrees is also shown to be inadequate for identifying the distribution of ISSRs, where in some seasons the majority of these are <100 km in width.
Although it is understood that these comments are pertinent to the AirTraf model, more justification of your assumptions for this model is needed if you are to propose it as a foundation for the current SolFinder decision making tool, which is weighing up routes based on at least one contrail avoidance objective function.

9) Line 79: "on" is not required.

10) Please explain briefly for clarity which atmospheric conditions are used.

11) Line 80: "on" is not required.

12) A detailed description of the method for obtaining fuel use is not given here. This is important given that BADA data does not rely on a physics based model, so you would need to explain why you used this in preference to Poll and Schumann's method as given in (Poll, An estimation method for the fuel burn and other performance characteristics of civil transport aircraft during cruise. Part 1: fundamental quantities and governing relations for a general atmosphere. 2021a, Poll, An estimation method for the fuel burn and other performance characteristics of civil transport aircraft during cruise. Part 2: determining the aircraft's characteristic parameters. 2021b). Although this is part of the AirTraf model, it becomes even more relevant when the results obtained using multiple objective functions which rely on atmospheric conditions are being compared, as is true for SolFinder. These conditions would inform the fuel use in the physics-based method.

13) Line 94: delete "s" of "facts" as singular is needed here.

14) A further explanation of why just one choice has been made for cost of fuel and time would be useful here. Presumably results will be affected by these weightings and so when discussing the relative merits of different solutions, these findings are dependent on pricing assumptions. Was any sensitivity analysis considered with different values?

15) Line 117: no need for capital F on future.

16) Line 139: "This allows to flexibly identify" should read "This allows flexible identification of".

17) It seems that only the flight position is being optimised here, neglecting the question of the airspeed. Previously AirTraf has been applied with flights at a constant Mach No., but airspeed has a bearing on fuel use and thus changes emissions too. As you are changing the temporal climate window every twelve minutes, the airspeed becomes more critical and some of the effects could be minimised more effectively by controlling both position and airspeed (C. K. Wells 2022, C. W. Wells 2023).

18) Line 152: Should Section A be Appendix A here for clarity? Also probably worth reiterating that the VIKOR algorithm is discussed fully in Appendix A in line 166 too and again in the caption for Figure 2. Given the importance of the VIKOR method in this system, I would prefer it to be at least described to some extent in the text and then this enlarged upon in Appendix A, but perhaps word count was an issue?

19) Line 152: "using as an example" reads better, you are missing the "an".

20) Line 162: Repetition of "follow" distracts from the start of the strategy description.

21) Section starting at line 175 is quite hard to follow, but appears important in justifying choice of parameters, so needs clarification.

22) Line 178: Should be "solutions" plural.

23) Line 194: "requires to" does not make sense here. Change to "wishes to" or "needs to"?

24) Line 203 includes an extra equation line label that needs removing.

25) In Figure 4 the red crosses are difficult to see. Perhaps a lighter green could be used for the surrounding markings?

26) Line 221: "it" is not needed here, "as is shown" suffices.

27) Line 222: "identify" is needed rather than "identified" here.

28) Line 230: Apostrophe missing in section title.

29) Figure 5 would be clearer if accompanied by a table displaying airport pairings.

30) Table 1: Again the choice of parameters is not fully justified, particularly the resolution, time step and choice of time period.

31) Line 265: Should read "Relative to our problem"

32) Figure 6: Do daily and monthly means have any real value here? If routes are different and days have different atmospheric conditions, averaging removes useful detail from the data. Can data be displayed to show results for each airport pair each day?

33) Line 294: "Pareto-fronts" here as you are using the plural.

34) Figure 9: Blue lines are too close in colour to differentiate in places. Could the colour choice be extended to make this diagram clearer?

35) Line 306: Short trajectories do not necessarily reduce fuel consumption, if you are mapping with respect to the ground and not the air. It is the air distance which allows for reduced fuel use.

36) Line 314: "which allow to minimize fuel use" should be "which allows minimal fuel use".

37) Line323: "resulting distributions from objective function values" reads better here.

38) Line 328: "of" is needed after independent, not "to".

39) In Figure 10 you specify flown distance, but not whether this is air distance or ground distance which, as mentioned before, is an important distinction.

40) The last part of Section 3.2 regards the relative effects of different emission factors as certain, whereas there is still much uncertainty in the actual radiative forcing

effects (Teoh 2020). This should be clarified here, rather than being left to a small paragraph at the end of the Discussion Section.

41) Line 345: "This study…10%" does not read properly. Please correct this.

42) Line 352: should read "than were found".

43) Line 356: This is the first mention of computational time.  Given your 12 minute time step, is this system running fast enough to allow the optimal trajectory under changing forecasts to be found from the multiple options calculated?  To justify the method, a discussion section on the timing of calculations is needed.

44) In Section 4 you should also be addressing:
   i.     Use of reanalysis data v. probabilistic data in a real time flight scenario.
   ii.    Computational time issues.
   iii.   Coarseness of the resolution when discussing contrail formation.
   iv.    Limitation of using just one month for atmospheric data.
   v.     Fuel burn being heavily dependent on airspeed and thus the limitation of using a single Mach number for the trajectories.

45) Line 386: "AirTraf…patterns" this is hard to follow and needs correcting. Splitting content between two sentences would make it far easier to express the meaning you are after.

46) Line 394: Can you explain this more clearly please? It sounds like you will concentrate your efforts on those flights where a significant emissions saving is possible and ignore the others, but please clarify potential levels of significance. This would also be a useful piece of analysis to include in the current paper.  A scatter graph of % climate improvement against % cost change, with coding to show what changes are considered significant at a 5% level in comparison with the flights actually flown each day would make the current research more relevant.

47) Line 416: "In fact", again use the singular here, not the plural. Also, "using $i_{best}$ as a reference point" reads better.

48) Line 417: Use "of" in place of "in" here. Place "a" in "such a reference point" and include "of the" between "each" and "objective".

The description of the VIKOR method is very clear in this appendix.

**Works Cited**

Poll, D. and Schumann, U. 2021a. "An estimation method for the fuel burn and other performance characteristics of civil transport aircraft during cruise. Part 1: fundamental quantities and governing relations for a general atmosphere." *The Aeronautical Journal* 125 (1284): 257-295.

Poll, D. and Schumann, U. 2021b. "An estimation method for the fuel burn and other performance characteristics of civil transport aircraft during cruise. Part 2: determining the aircraft's characteristic parameters." *The Aeronautical Journal* 125 (1284): 296-340.

Reutter, P., Neis, P., Rohs, S. and Sauvage, B. 2020. "Ice supersaturated regions: properties and validation of ERA-Interim reanalysis with IAGOS in situ water vapour measurements." *Atmospheric Chemistry and Physics* 20 (2): 787-804.

Teoh, R., Schumann, U., Majumdar, A. and Stettler, M. 2020. "Mitigating the Climate Forcing of Aircraft Contrails by Small-Scale Diversions and Technology Adoption." *Environmental science technology* 54 (5): 2941-2950.

Wells, C., Kalise, D., Nichols, N., Poll, D. and Williams, P. 2022. "The role of airspeed variability in fixed time, fuel-optimal aircraft trajectory planning." *Optimization and Engineering.*

Wells, C., Williams, P., Nichols, N., Kalise, D. and Poll, D. 2023. "Minimising emissions from flights through realistic wind fields with varying aircraft weights." *Transportation Research Part D: Transport and Environment* 117.

---

## Author Comment (AC1)

**Author's response to Referee #1 (gmd-2023-88)**

We would like to thank referee #1 for the constructive comments on the manuscript we submitted. We took all comments into account. In this document, we include all comments from the referee (indicated in *italic*) and we provide a point-by-point response, also describing the changes made in the manuscript.

**Referee:** *The paper is relevant to the area of aircraft routing and addresses the choice between multiple optimal routes dependent on a range of objectives. As such this is not a major change to routing strategies, but rather a next step in an evolving process. This means that the novelty of the approach lies in applying an established algorithm to a new setting. The advance this affords allows the previously defined AirTraf model to be used in a different and more integrated way. The method described is mostly clear, but some of the assumptions supporting the use of AirTraf in contrail avoidance are not properly justified, given research results previously published on the nature of super saturated icy regions. This is further explored in the more detailed file that is attached below.*

*Results are somewhat limited by the use of a single month, which when looking at climatic conditions provides a narrower range of possible variable value combinations than is most useful. However, conclusions across this reduced timeframe given the other model assumptions are supported by the research that has been completed. Given the reliance of the results on a combination of different models and the limited explanation in this paper of the climate inputs and methods for fuel use calculation, it would be difficult to reproduce results from this work alone, but taken alongside previous research and given the limited access to models available, replication of some of the results could be possible.*

*The work is properly referenced and the need for the model is justified, with the paper title including all necessary detail. The abstract is concise and reflects the paper content, but could be better worded (see attached comments). The main structure of the paper is good, but there are occasions where order within sections would be improved by small changes. The language and grammar of the paper need minor corrections, which have been noted in the attached comments. Clear definitions of formulae, symbols and abbreviations are, however, all present in the paper.*

*Overall my recommendation is for the paper to be accepted subject to minor revisions. Where assumptions weaken the usefulness of the model, this stems more from the original AirTraf usage than the current decision-making tool, so whilst these choices do need more justification in this paper, they are not grounds for major revisions.*

**Author's response:** We thank the referee for this feedback. In the revised manuscript, we added new passages discussing the assumptions taken in this study, in particular relatively to the identification of ice supersaturated regions, the atmospheric conditions received as input by the model, and the methods for the fuel use calculation. Addressing the detailed comments listed below, we believe that the methodology is now described in a more complete and clear way, which would assist in replicating our results. We pay particular attention to explaining the assumptions taken in the calculation of the objective functions within the AirTraf model. Lastly, the quality of the text improved by including the grammar and language corrections highlighted by the referee.

**Point-by-point responses**

1. *Abstract: lines 4-6 "This paper…solutions". This phrase does not read properly. Suggest: "…which allow the reduction of the flights'…"*
   **Author's response:** This sentence has been rephrased as suggested.

2. *Line 14: "to 3-5% of the total", the "to" and the "the" are not required.*
   **Author's response:** We corrected this passage.

3. *Line 26: None of the references that have been included alluded to work on minimising just carbon dioxide emissions. Given the uncertainty of non-carbon dioxide effects, which is not alluded to until the Discussion Section, this should be mentioned here. There is also no justification for the use of strategic rather than tactical planning for avoidance of ice super saturated regions (ISSR) to prevent contrail formation, which often cannot be accurately forecast pre-flight (Reutter 2020).*
   **Author's response:** In line 26, we refer to studies that explored the possibility of using the time and space dependency of non-$CO_2$ effects to reduce the climate impact of aviation via aircraft trajectory optimization. As mentioned at line 19, the "temperature perturbation resulting from $CO_2$ emissions is only dependent on the amount of emitted $CO_2$, due to the long atmospheric lifetime of $CO_2$". Therefore, we follow this line with a new passage that includes strategies aiming at reducing only $CO_2$ effects, which also contributes to clarify the difference between $CO_2$ and non-$CO_2$ effects. We also agree with referee #1 on the importance of including here the difference in the current confidence levels of $CO_2$ and non-$CO_2$ effects estimates, as well as the main advantages and disadvantages of strategic and tactical planning. Therefore, a paragraph has been added to the Introduction, to address these points.

4. *Lines 35-42: These would read better if requirements and solutions were given together, rather than listing all requirements and then all solutions. A list structure would also make the arguments clearer, rather having numbered points lost in a passage of prose.*
   **Author's response:** Thank you for underlying this passage. As the different requirements do not correspond to specific solutions, but rather to the SolFinder module, we highlight them using a list structure as suggested, without numbers, while the solutions are kept as numbered points.

5. *Line 45: "This modelling chain allows to select…" does not make sense as it stands. Allows who to select?*
   **Author's response:** We rephrased this sentence as: "This modelling chain enables users to select…".

6. *Line 68: Space needed after the comma.*
   **Author's response:** The space has been added.

7. *Line 73-76: This description is very unclear. What do you mean by nudging? Please explain in more detail exactly which climate indices are used and how these are processed.*
   **Author's response:** "Nudging" is a technique that is widely used to align the atmospheric conditions simulated by a model to those observed during a specific period of time. A list of the variables used for nudging is given in the manuscript at line 73-76 (i.e.: divergence,

vorticity, temperature and the (logarithm of the) surface pressure). In the revised version of the paper, we clarify the passage by splitting the sentence into two parts, and adding more details on how the reanalysis data is assimilated by the model.

8. *Section 2.1: Using reanalysis data does not take into account the difficulty in forecasting ISSR, which should be acknowledged. Seasonal patterns in ISSR formation are also not included when a single month is considered. In addition, Reutter et al (2020) show that ERA Interim reanalysis data misrepresents ice supersaturation at flight altitudes and that there is a significantly larger fraction of ISSR in measurements of water vapour. A resolution of 2.8 degrees is also shown to be inadequate for identifying the distribution of ISSRs, where in some seasons the majority of these are <100 km in width. Although it is understood that these comments are pertinent to the AirTraf model, more justification of your assumptions for this model is needed if you are to propose it as a foundation for the current SolFinder decision making tool, which is weighing up routes based on at least one contrail avoidance objective function.*
   **Author's response:** Forecasting ISSR has been identified as one of the main challenges towards the implementation of contrail avoidance strategies [Molloy et al., 2022]. We include this point in the Introduction section of the revised paper, as part of the overview on the challenges linked to the implementation of climate-optimized aircraft trajectories. In the present study, we focused on one month of simulation to demonstrate the usage of SolFinder coupled to AirTraf, thus the seasonal patterns (for example in ISSRs formation) are not captured. However, the developed modelling chain allows to extend the analysis to a larger number of weather conditions. Therefore, in the next phase of the research we will analyse seasonal variabilities to "identify those weather situations allowing for the largest reductions in the temperature response from aviation emissions via the optimization of aircraft trajectories" (lines 397-398 of the paper draft). Moreover, to account for the horizontal resolution of the model of 2.8 degrees, a parameterization has been developed by Burkhardt et al., (2008) to estimate the fraction of model grid box which is supporting persistent contrails. With this method, which is implemented in the EMAC submodel CONTRAIL [Frömming et al., 2014], we consider that the majority of ISSRs have characteristic dimensions that are smaller than the model resolution.

9. *Line 79: "on" is not required.*
   **Author's response:** We removed "on" as suggested.

10. *Please explain briefly for clarity which atmospheric conditions are used.*
    **Author's response:** The AirTraf submodel requires as input atmospheric conditions, such as wind and temperature fields, to determine the fuel consumption and emissions of aircraft trajectories [Yamashita et al, 2016]. Indirectly, the submodel is also strongly reliant on the variables provided to the ACCF submodel (temperature, potential vorticity, relative humidity, outgoing longwave radiation, … ) for the estimation of the Average Temperature Response over 20 years (ATR20) of non-$CO_2$ effects of a specific flight, which are used by AirTraf to calculate the objective function of climate-optimized trajectories [Yin et al., 2023]. This input is now briefly explained at line 71 of the revised paper.

11. *Line 80: "on" is not required.*
    **Author's response:** We removed "on" as suggested.

12. *A detailed description of the method for obtaining fuel use is not given here. This is important given that BADA data does not rely on a physics based model, so you would need to explain why you used this in preference to Poll and Schumann's method as given in:*

   - *Poll, An estimation method for the fuel burn and other performance characteristics of civil transport aircraft during cruise. Part 1: fundamental quantities and governing relations for a general atmosphere. 2021a.*
   - *Poll, An estimation method for the fuel burn and other performance characteristics of civil transport aircraft during cruise. Part 2: determining the aircraft's characteristic parameters. 2021b.*

   *Although this is part of the AirTraf model, it becomes even more relevant when the results obtained using multiple objective functions which rely on atmospheric conditions are being compared, as is true for SolFinder. These conditions would inform the fuel use in the physics-based method.*
   **Author's response:** Thank you for highlighting this point. In the reviewed version of the paper, we include a brief description of the method used to calculate the fuel used (Section 2.2). AirTraf employs a total energy model, which is based on the BADA methodology and the DLR fuel flow method (H. Yamashita et al., 2016). Current research is exploring the possibility of offering alternative, open-source aircraft performance models, such as OpenAP (https://openap.dev/), or the Poll and Schumann's method referenced by referee #1.

13. *Line 94: delete "s" of "facts" as singular is needed here.*
   **Author's response:** The "s" has been removed.

14. *A further explanation of why just one choice has been made for cost of fuel and time would be useful here. Presumably results will be affected by these weightings and so when discussing the relative merits of different solutions, these findings are dependent on pricing assumptions. Was any sensitivity analysis considered with different values?*
   **Author's response:** We are aware of the complexity of calculating realistic operating costs. In this study, we use a simplified representation of the operating cost, the simple operating costs (SOC) defined in Eq. (1), in order to include both fuel use and flight time in the optimization process. Our results are affected by the choice of the cost of fuel, which is affected by fluctuations in time, and by assuming a linear relationship between cost of time and time of flight, thus neglecting economic penalties caused by delays. In this study, we do not intend to evaluate the sensitivity of our results due to these assumptions, as the focus is to illustrate how the SolFinder module can be employed to identify compromise solutions between the optimization objectives defined in Eq. (1) and Eq. (2). Current research is being conducted using different values of these parameters, for example, employing more recent values of the cost of fuel.

15. *Line 117: no need for capital F on future.*
   **Author's response:** We keep the capital F on future, to indicate how it corresponds to the F in the acronym F-ATR20. This method is chosen throughout the paper to define all acronyms (for example, ATR20 at line 109).

16. *Line 139: "This allows to flexibly identify" should read "This allows flexible identification of".*
   **Author's response:** The suggested change has been adopted.

17. *It seems that only the flight position is being optimised here, neglecting the question of the airspeed. Previously AirTraf has been applied with flights at a constant Mach No., but airspeed has a bearing on fuel use and thus changes emissions too. As you are changing the temporal climate window every twelve minutes, the airspeed becomes more critical and some of the effects could be minimised more effectively by controlling both position and airspeed (C. K. Wells 2022, C. W. Wells 2023).*
**Author's response:** Thank you for pointing this out. Indeed, here the optimization is only performed with respect to the flight position (11 design variables describing eight control points, see Section 2.2), while keeping a constant Mach number. Current work is addressing this limitation, thus we plan to include airspeed as an additional design variable in a future version of AirTraf.

18. *Line 152: Should Section A be Appendix A here for clarity? Also probably worth reiterating that the VIKOR algorithm is discussed fully in Appendix A in line 166 too and again in the caption for Figure 2. Given the importance of the VIKOR method in this system, I would prefer it to be at least described to some extent in the text and then this enlarged upon in Appendix A, but perhaps word count was an issue?*
**Author's response:** We agree with referee #1 on the opportunity of including a brief description of the VIKOR method in Section 2.3.1, while referring to Appendix A for more details. Therefore, the sentence at line 152 has been extended. We also added references to Appendix A at line 166 and in the caption of Figure 2.

19. *Line 152: "using as an example" reads better, you are missing the "an".*
**Author's response:** We adopted the suggested change.

20. *Line 162: Repetition of "follow" distracts from the start of the strategy description.*
**Author's response:** We changed "following steps" with "steps listed here".

21. *Section starting at line 175 is quite hard to follow, but appears important in justifying choice of parameters, so needs clarification.*
**Author's response:** We rephrased this section to clarify its message. In particular, we separated more clearly the impact of choosing different values of the relative weights of the objectives (w), to the impact of using different values of the group utility weight (γ).

22. *Line 178: Should be "solutions" plural.*
**Author's response:** This typo has been corrected.

23. *Line 194: "requires to" does not make sense here. Change to "wishes to" or "needs to"?*
**Author's response:** Changed to "wishes to".

24. *Line 203 includes an extra equation line label that needs removing.*
**Author's response:** We removed the extra equation line label.

25. *In Figure 4 the red crosses are difficult to see. Perhaps a lighter green could be used for the surrounding markings?*

**Author's response:** Figure 4 has been modified to increase the visibility of the red crosses, as shown below.

[Figure]

*Figure 4 - Example of selecting the solution among the Pareto-surface matching a target increase in 0.5% in flight time (indicated by red triangles). The green dots indicate the Pareto-optimal solutions, which result from a tri-objective optimization problem minimizing flight time, fuel use, and ATR20_tot.*

26. *Line 221: "it" is not needed here, "as is shown" suffices.*
    **Author's response:** We removed "it".

27. *Line 222: "identify" is needed rather than "identified" here.*
    **Author's response:** We corrected this typo.

28. *Line 230: Apostrophe missing in section title.*
    **Author's response:** The new section title is: "Application of decision-making method to analyse trajectories' variability along Pareto-front"

29. *Figure 5 would be clearer if accompanied by a table displaying airport pairings.*
    **Author's response:** We added Appendix B, with the list of airport pairings included in our air traffic sample (included below).

| | ICAO airport code | | | | ICAO airport code | |
|---|---|---|---|---|---|---|
| citypairs | departure | arrival | citypairs | departure | arrival |
| 0 | LTFM | EGLL | 50 | EDDL | LTFM |
| 1 | EGLL | LTFM | 51 | LTFM | EDDL |
| 2 | LEMD | GCLP | 52 | LFPO | LFBO |
| 3 | GCLP | LEMD | 53 | LEMD | EHAM |
| 4 | GCXO | LEMD | 54 | LFBO | LFPO |
| 5 | LEMD | GCXO | 55 | EHAM | LEMD |
| 6 | LFPG | LTFM | 56 | LFMN | LFPO |
| 7 | LTFM | LFPG | 57 | LFPO | LFMN |
| 8 | EGLL | LEMD | 58 | EGLL | LPPT |
| 9 | LFPO | LPPT | 59 | LPPT | EGLL |
| 10 | LPPT | LFPO | 60 | LEBL | LEMD |
| 11 | LEMD | EGLL | 61 | LFPG | LIRF |
| 12 | LEPA | EDDL | 62 | LIRF | LFPG |
| 13 | EDDL | LEPA | 63 | EHAM | LIRF |
| 14 | EHAM | LTFM | 64 | LIRF | EHAM |
| 15 | LTFM | EHAM | 65 | LEPA | EDDT |
| 16 | LGAV | EGLL | 66 | EDDT | LEPA |
| 17 | EGLL | LGAV | 67 | LEMD | LFPO |
| 18 | LEBL | EGKK | 68 | LFPO | LEMD |
| 19 | EGKK | LEBL | 69 | EDDM | EGLL |
| 20 | LEMD | LIRF | 70 | EGLL | EDDM |
| 21 | LIRF | LEMD | 71 | LEMD | LEBL |
| 22 | ESSA | EGLL | 72 | UBBB | LTFM |
| 23 | EGLL | ESSA | 73 | LTCG | LTFJ |
| 24 | EHAM | LEBL | 74 | LTFJ | LTCG |
| 25 | LEBL | EHAM | 75 | LFPG | LGAV |
| 26 | EDDF | LEMD | 76 | LGAV | LFPG |
| 27 | EGCC | GCTS | 77 | LEMD | EDDM |
| 28 | LEMD | EDDF | 78 | EDDM | LEMD |
| 29 | GCTS | EGCC | 79 | ESSA | LEMG |
| 30 | LPPT | EDDF | 80 | LEMG | ESSA |
| 31 | EDDF | LPPT | 81 | LEMD | EBBR |
| 32 | LIRF | EGLL | 82 | EBBR | LEMD |
| 33 | EGLL | LIRF | 83 | LROP | EGGW |
| 34 | LPPT | EHAM | 84 | EGGW | LROP |
| 35 | EHAM | LPPT | 85 | LPPT | EBBR |
| 36 | ENGM | ENTC | 86 | LTFM | UBBB |
| 37 | LTFM | EDDF | 87 | EBBR | LPPT |
| 38 | EDDF | LTFM | 88 | LEMG | EFHK |
| 39 | EGKK | LEMG | 89 | EFHK | LEMG |
| 40 | ENTC | ENGM | 90 | GCXO | LEBL |
| 41 | LEMG | EGKK | 91 | LEBL | GCXO |
| 42 | EGLL | EFHK | 92 | LTAI | EDDK |
| 43 | EFHK | EGLL | 93 | EDDK | LEPA |
| 44 | LTAI | EDDL | 94 | LEPA | EDDK |
| 45 | EDDL | LTAI | 95 | EDDH | LEPA |
| 46 | GCTS | EGKK | 96 | EDDT | EDDF |
| 47 | EGKK | GCTS | 97 | LEPA | EDDH |
| 48 | LEMG | EKCH | 98 | EDDF | EDDT |
| 49 | EKCH | LEMG | 99 | LPPR | LFPO |

*Figure B1-  List of origin/destination airport pairs included in the air traffic sample.*

**30.** *Table 1: Again the choice of parameters is not fully justified, particularly the resolution, time step and choice of time period.*
**Author's response:** Both the temporal and spatial resolutions are chosen as those used in the benchmark tests conducted in Yamashita et. al. (2016), in which the parameterization of the genetic algorithm was determined. Relatively to the time period, a simulation of the duration of one month was chosen to illustrate the capability of the model to solve trajectory optimization problems over consecutive days, rather than representative weather patterns, as was the case in previous studies (e.g., Grewe et al., 2017). In particular, the year 2018 was chosen because of the availability of an European air traffic sample based on the Available Seat Kilometres (ASK) for the European Civil Aviation Conference (ECAC) area in 2018. A winter month (January) was chosen because, to obtain ATR20 estimates, we rely on the aCCFs version 1.0A, which were developed using representative weather patterns in winter and summer, thus it is recommended to employ them for the atmospheric conditions occurring during these seasons [Dietmüller et al., 2023].

**31.** *Line 265: Should read "Relative to our problem"*
**Author's response:** We changed "Relatively" to "Relative".

**32.** *Figure 6: Do daily and monthly means have any real value here? If routes are different and days have different atmospheric conditions, averaging removes useful detail from the data. Can data be displayed to show results for each airport pair each day?*
**Author's response:** The daily and monthly means in Figure 6 provide an indication of the mitigation potential of climate-optimal and compromise solutions with respect to cost-optimal solutions, over the considered temporal and spatial domain. The aggregated results can also be more easily compared with previous studies, e.g., Grewe et al. 2017 (see Discussion section). The daily values are also showed in figure 6, to highlight how the relationship between the relative changes in climate impact, $\Delta ATR20_{tot}$ [%], and in simple operating cost, $\Delta SOC$ [%], varies depending on the atmospheric conditions. Therefore, these averaged values are useful, and we keep them in the revised version of the paper. However, we agree that meaningful information on the variability over single routes, as we are ultimately interested in specific solutions yielding low cost increase and high climate impact reduction. Therefore, we add a panel in the updated Figure 6 (see below) with the aim of showing the full variability of the data.

[Figure]

*Figure 6 - Relation between the relative changes in climate impact, ΔATR20tot [%], and in simple operating cost, ΔSOC [%], with respect to the cost-optimal solution. Panel a): scatter graph of ΔATR20tot [%] against ΔSOC [%], comparing the values obtained varying the weight of simple operating costs $w_{SOC}$. Panel b): values obtained summing over the 100 routes optimized per day. The black line illustrates the average values over the 31 days included in the simulations, connecting the points selected varying the VIKOR weight $w_{SOC}$ from 0.2 to 0.9 (green dots). The extremes of the Pareto fronts (climate- and cost-optimal solutions, red dots) are included. The gray lines represent the Pareto fronts obtained on each simulation day.*

**33.** *Line 294: "Pareto-fronts" here as you are using the plural.*
    **Author's response:** This typo has been corrected.

**34.** *Figure 9: Blue lines are too close in colour to differentiate in places. Could the colour choice be extended to make this diagram clearer?*
    **Author's response:** Figure 9 has been modified to improve its readability. In the updated figure, the style of the lines have been changed, so that curves drawn with two consecutive colours can be better compared by distinguishing their style.

[Figure]

*Figure 9 - Relative frequencies [%] of different values of the climate-cost coefficient k [$/K], comparing the SolFinder solution-picking strategies using VIKOR (blue curves) or the target SOC change (red curves). The curves approximate the histogram outlines (connecting the bars centres) to highlight the shapes of the distributions and facilitate their comparison. Each curve includes the values obtained with different decision-making strategies, considering the 100 flights optimized on each simulation day (31\*100 values per histogram).*

35. *Line 306: Short trajectories do not necessarily reduce fuel consumption, if you are mapping with respect to the ground and not the air. It is the air distance which allows for reduced fuel use.*
    **Author's response:** Thank you for noticing this. Indeed, the great circle option from AirTraf would provide the shortest trajectory, which may differ from the cost-optimal trajectory. We rephrased this as: "cost-optimal flights are characterized by the highest mean flight altitudes and the shortest trajectories among the solutions considered, due to the presence of fuel consumption in the optimization objective."

36. *Line 314: "which allow to minimize fuel use" should be "which allows minimal fuel use".*
    **Author's response:** This sentence has been rephrased.

37. *Line323: "resulting distributions from objective function values" reads better here.*
    **Author's response:** We adopted the suggested change.

38. *Line 328: "of" is needed after independent, not "to".*
    **Author's response:** Now "of" is used after independent.

39. *In Figure 10 you specify flown distance, but not whether this is air distance or ground distance which, as mentioned before, is an important distinction.*
    **Author's response:** The label in Figure 10 has been changed to "ground distance".

40. *The last part of Section 3.2 regards the relative effects of different emission factors as certain, whereas there is still much uncertainty in the actual radiative forcing effects (Teoh 2020). This should be clarified here, rather than being left to a small paragraph at the end of the Discussion Section.*

**Author's response:** We agree on the importance of highlighting the uncertainties affecting non-CO$_2$ effects estimates when describing our results in Section 3.2. Indeed, this issue is introduced already in the Introduction of the revised version of the paper, where we explain how the current level of scientific understanding of the non-CO$_2$ effects of aviation is lower than the one of CO$_2$ effects, as demonstrated by the uncertainty ranges of the radiative forcing estimates reported by Lee et al. (2021). Therefore, the relative importance of the different effects of aviation emission can vary, depending on the regions of the uncertainty range considered. We include this point when discussing our results, as suggested by the referee.

41. *Line 345: "This study…10%" does not read properly. Please correct this.*
    **Author's response:** We replaced this sentence with "This study found that a 10% reduction in climate impact can be achieved with a 1.0% increase in operating costs".

42. *Line 352: should read "than were found".*
    **Author's response:** We replaced "than what we found" with "than were found".

43. *Line 356: This is the first mention of computational time. Given your 12 minute time step, is this system running fast enough to allow the optimal trajectory under changing forecasts to be found from the multiple options calculated? To justify the method, a discussion section on the timing of calculations is needed.*
    **Author's response:** If we correctly interpret this comment, we observe a misunderstanding of our methodology. Therefore, here we summarize the different timings involved in our experiments, while being available for further clarifications in case this response does not fully cover the referee's remark. AirTraf receives as input atmospheric conditions with a time resolution of 12 minutes. At the beginning of every time step, AirTraf checks if the time of departure of a flight in the air traffic sample has been reached. If this is the case, the trajectory is calculated and optimized according to the selected strategy. When this step is completed, the model proceeds with the following time step. At each subsequent timestep the aircraft moves according to the trajectory identified at the time of departure, and the flight properties are calculated according to the local atmospheric conditions. The flying process of AirTraf is illustrated in more detail in Figure 3 of Yamashita et al. (2016). This modelling chain has been developed for the purpose of evaluating the mitigation potential of climate-optimized trajectories under a large number of weather conditions (e.g., on every day over 1-3 years). Using computing resources provided by the TU Delft High Performance Cluster (HPC12), a simulation optimizing 100 flights per day over a full year is completed in less than a month. Therefore, this system can be used to assess the diurnal and seasonal variability of the mitigation potential of eco-efficient trajectories, and to identify patterns over different routes/geographic areas.

44. *In Section 4 you should also be addressing:*

    *i. Use of reanalysis data v. probabilistic data in a real time flight scenario.*

    *ii. Computational time issues.*

    *iii. Coarseness of the resolution when discussing contrail formation.*

    *iv. Limitation of using just one month for atmospheric data.*

> *v. Fuel burn being heavily dependent on airspeed and thus the limitation of using a single Mach number for the trajectories.*

**Author's response:** These points were added to the discussion in Section 4, as described in previous bullet points. In particular:

> i. At point 8: use of reanalysis data v. probabilistic data in a real time flight scenario.

> ii. At point 43: Computational time issues.

> iii. At point 8: Coarseness of the resolution when discussing contrail formation.

> iv. At point 8: Limitation of using just one month for atmospheric data.

> v. At point 17: Fuel burn being heavily dependent on airspeed and thus the limitation of using a single Mach number for the trajectories.

45. *Line 386: "AirTraf…patterns" this is hard to follow and needs correcting. Splitting content between two sentences would make it far easier to express the meaning you are after.*
**Author's response:** We replaced the sentence at line 386 with the following passage: " We showed here how the selected decision-making strategies can be used to identify solutions matching specific preferences (e.g., eco-efficient aircraft trajectories). Moreover, using this modelling chain, it is possible to explore the results variability under a large number of consecutive days, due to the coupling between SolFinder and an atmospheric chemistry model (EMAC), via the EMAC submodel AirTraf."

46. *Line 394: Can you explain this more clearly please? It sounds like you will concentrate your efforts on those flights where a significant emissions saving is possible and ignore the others, but please clarify potential levels of significance. This would also be a useful piece of analysis to include in the current paper. A scatter graph of % climate improvement against % cost change, with coding to show what changes are considered significant at a 5% level in comparison with the flights actually flown each day would make the current research more relevant.*
**Author's response:** In the passage at line 394 we indeed anticipate that, in future research, we will focus on those flights leading to the largest climate impact reductions, comparing the various optimized flights. This will differ from the approach presented in this paper, where single-flight optimization is performed, and the most eco-efficient solution is identified considering each flight independently from the others. We interpreted this comment as an incentive of defining a threshold of significance for the relative changes in operating cost and climate impact. We addressed this remark as indicated in the following, but in case of misunderstanding, we can provide further clarification. As described in the bullet point 32., we added a panel in Figure 6 with a scatter graph of the relative reduction in climate impact, against the relative change in operating cost, using as reference scenario our cost-optimal solutions. Depending on the constraints, this reference scenario affects the resulting climate impact mitigation potential. The impact of using as baseline real flights has been explored during the FlyATM4E project[1] for a limited number of flights. Moreover, we are aware that airspace structure and capacity could limit the mitigation potential of implementing this operational strategy. Future research will address these factors, but we cannot currently include these considerations in the present paper.
* * *
[1] See section 3.2 of Deliverable 3.2 at https://flyatm4e.eu/deliverables/ .

47. *Line 416: "In fact", again use the singular here, not the plural. Also, "using ibest as a reference point" reads better.*
**Author's response:** We adopted the suggested changes.

48. *Line 417: Use "of" in place of "in" here. Place "a" in "such a reference point" and include "of the" between "each" and "objective". The description of the VIKOR method is very clear in this appendix.*
**Author's response:** We corrected this part as suggested. Thank you for this positive remark.

**Works Cited**

Burkhardt, U., Kärcher, B., Ponater, M., Gierens, K., and Gettelman, A.: Contrail cirrus supporting areas in model and observations, Geo-physical Research Letters, 35, https://doi.org/10.1029/2008GL034056, 2008.

Dietmüller, S., Matthes, S., Dahlmann, K., Yamashita, H., Simorgh, A., Soler, M., Linke, F., Lührs, B., Meuser, M. M., Weder, C., Grewe, V., Yin, F., Castino, F., and Dietmüller, S.: A python library for computing individual and merged non-CO 2 algorithmic climate change functions: CLIMaCCF V1.0, 2023.

Frömming, C., Grewe, V., Brinkop, S., and Jöckel, P.: Documentation of the EMAC submodels AIRTRAC 1.0 and CONTRAIL 1.0, supplementary material of Grewe et al., 2014, 7, 175–201, Geoscientific Model Development, https://doi.org/10.5194/gmd-7-175-2014%0A, 2014.

Grewe, V., Matthes, S., Frömming, C., Brinkop, S., Jöckel, P., Gierens, K., Champougny, T., Fuglestvedt, J., Haslerud, A., Irvine, E., and Shine, K.: Feasibility of climate-optimized air traffic routing for trans-Atlantic flights, Environmental Research Letters, 12, https://doi.org/10.1088/1748-9326/aa5ba0, 2017.

Molloy, J.; Teoh, R.; Harty, S.; Koudis, G.; Schumann, U.; Poll, I.; Stettler, M.E.J. Design Principles for a Contrail-Minimizing Trial in the North Atlantic. Aerospace 2022, 9, 375. https://doi.org/10.3390/aerospace9070375

Poll, D. and Schumann, U. 2021a. "An estimation method for the fuel burn and other performance characteristics of civil transport aircraft during cruise. Part 1: fundamental quantities and governing relations for a general atmosphere." The Aeronautical Journal 125 (1284): 257-295.

Poll, D. and Schumann, U. 2021b. "An estimation method for the fuel burn and other performance characteristics of civil transport aircraft during cruise. Part 2: determining the aircraft's characteristic parameters." The Aeronautical Journal 125 (1284): 296-340.

Reutter, P., Neis, P., Rohs, S. and Sauvage, B. 2020. "Ice supersaturated regions: properties and validation of ERA-Interim reanalysis with IAGOS in situ water vapour measurements." Atmospheric Chemistry and Physics 20 (2): 787-804.

Teoh, R., Schumann, U., Majumdar, A. and Stettler, M. 2020. "Mitigating the Climate Forcing of Aircraft Contrails by Small-Scale Diversions and Technology Adoption." Environmental science technology 54 (5): 2941-2950.

Wells, C., Kalise, D., Nichols, N., Poll, D. and Williams, P. 2022. "The role of airspeed variability in fixed time, fuel-optimal aircraft trajectory planning." Optimization and Engineering.

Wells, C., Williams, P., Nichols, N., Kalise, D. and Poll, D. 2023. "Minimising emissions from flights through realistic wind fields with varying aircraft weights." Transportation Research Part D: Transport and Environment 117

Yamashita, H., Grewe, V., Jöckel, P., Linke, F., Schaefer, M., and Sasaki, D.: Air traffic simulation in chemistry-climate model EMAC 2.41: AirTraf 1.0, Geoscientific Model Development, 9, 3363–3392, https://doi.org/10.5194/gmd-9-3363-2016, 2016.

Yin, F., Grewe, V., Castino, F., Rao, P., Matthes, S., Dahlmann, K., Dietmüller, S., Frömming, C., Yamashita, H., Peter, P., Klingaman, E., Shine, K. P., Lührs, B., and Linke, F.: Predicting the climate impact of aviation for en-route emissions: the algorithmic climate change function submodel ACCF 1.0 of EMAC 2.53, Geoscientific Model Development, 16, 3313–3334, https://doi.org/10.5194/gmd-16-3313-2023, 2023.

---

## Author Comment (AC2)

**Author's response to Referee #2 (gmd-2023-88)**

We are grateful to referee #2 for the positive and constructive feedback. Below, we list all comments from the referee (in *italic*) and our point-by-point response.

**Referee:** *This manuscript presents the SolFinder tool, a multi-criteria decision making (MCDM) application designed to identify eco-efficient aircraft trajectories. It accomplishes this by solving a bi-objective optimization problem aimed at minimizing both climate impact and operating costs. The development of the MCDM tool, along with the results it yields, appears to be of interest to the readership of Geoscientific Model Development. Overall, the manuscript is clearly written and generally well-structured. I recommend only minor revisions to address the points listed below before accepting it for publication.*

1. *The terms "Pareto-optimal solutions" and "Pareto optimal solutions" are both accepted in the literature; however, consistency within your manuscript would enhance its presentation. Please choose one format and use it uniformly throughout the paper.*
   **Author's response:** Thank you for pointing this out, we adopted "Pareto-optimal solutions" consistently in the revised version of the manuscript.

2. *On line 156, the authors write, "If the VIKOR method identifies more than one recommended solution, i.e., the solutions $p_v$ (v = 1,2,...,M) are equally recommended, the model selects the one with the minimum value of the objective function assigned to the lowest weight $w_n$", justify why it opts for the solution with the minimum value of the objective function corresponding to the lowest weight. Would it be more logical for the final selection to be based on the objective with the highest weight since it represents the most important criterion?*
   **Author's response:** We thank the referee for highlighting the need of clarifying this point. This criterion was chosen so that, in the following situation:
   - bi-objective optimization problem, aiming to minimize operating costs and climate impact
   - weight assigned to operating costs is higher than weight assigned to climate impact

   we find a way to translate in mathematical terms our definition of "eco-efficient" aircraft trajectories, i.e., a compromise solution between cost-optimal and climate-optimal solutions, such that the largest possible climate impact reduction is achieved, while keeping the operating costs nearly unchanged with respect to the cost-optimal solution. Using the VIKOR method, a subset of Pareto-optimal solutions is identified, according to the relative importance of the two optimization objectives. Therefore, if the highest weight is assigned to the objective function representing operating costs, the VIKOR method equally recommends a subset of Pareto-optimal solutions close to – or, possibly, including - the cost-optimal extreme point of the Pareto-front. Among this subset of equally recommended solutions, we choose the point leading to the largest climate impact reduction, i.e., "the minimum value of the objective function assigned to the lowest weight" (line 156). Therefore, the objective with the highest weight plays a dominant role in the selection of the subset of equally recommended solutions (VIKOR method), while the objective with the lowest weight becomes dominant in the selection of a single solution among this subset. We clarify this motivation in the revised version of the manuscript, with the support of Figure 2.

3. *Figure 2 could be enhanced for clarity and visual appeal. Firstly, it would be beneficial to use a single color and symbol to denote the Pareto-optimal solutions—currently, there is a mix of*

*red dots and gray crosses, which can be confusing. Secondly, adopt a consistent color and symbol for the solutions recommended by VIKOR, which are presently indicated by both red dots and green crosses. Lastly, I suggest using a distinctly different color and symbol to highlight the final selected solution in Figure 2d. A blue star or another distinctive marker could be effective in clearly indicating the chosen solution to the audience.*

**Author's response:** We agree with the suggestion of improving the readability of Figure 2. Below, we include the updated figure, using consistent indicators for the different solutions (Pareto-optimal solutions, solutions selected by VIKOR, final selected solution) across the four panels. Symbols and colours used for different categories of optimal solutions are now also consistent between Fig. 2 and Fig. 3.

[Figure]

*Figure 2 - Illustration of the steps performed by the eco-efficient decision-making strategy relying of VIKOR. The aircraft trajectories are optimized to minimize SOC and ATR20, resulting in a set of Pareto-optimal solutions (grey crosses). We set $w_{SOC} = 0.7$, $w_{ATR20} = 0.3$, $\gamma = 0.5$. Panel a) shows the Pareto-optimal solutions (grey crosses) collected before applying the decision-making strategy. Panel b) illustrates the application of the VIKOR method, thus the axes are scaled as in Fig. A1. This*

*step results in the identification of the subset of recommended solutions, represented by the green triangles in panel c). Panel d) shows the selected solution (red dot) among the subset of recommended solutions (green triangles).*

4. *Ensure consistency in your manuscript by using either "decision-maker" or "decision maker" throughout the paper.*
   **Author's response:** Thank you for highlighting this inconsistency. In the revised manuscript, we only use the option "decision-maker".

5. *The manuscript mentions that decision-makers must configure the values for gamma (γ) and weights when using the VIKOR method implemented within SolFinder. While determining weights may be more intuitive for decision-makers, selecting an appropriate gamma (γ) value could pose a challenge for those less familiar with the subject of the MCDM or VIKOR method. I recommend that the authors include a brief, accessible explanation of how varying gamma (γ) values influence the outcomes. For example, elucidate the implications of a gamma (γ) value near 1 versus one closer to 0, and clearly state the default gamma value used in your program, such as 0.5, if applicable. This explanation would benefit both the paper and the SolFinder tool itself, leveraging insights from your good sensitivity analyses.*
   **Author's response:** The impact of using different values of the parameter gamma (γ) has been addressed in the revised version of the manuscript, by extending the text in the "Sensitivity of VIKOR parameterization" section (Sect. 2.3.1.). We use Figure 3 to explain how, as explained in Opricovic and Tzeng (2004):
   - with $\gamma < 0.5$, the veto principle is applied, i.e., if one of the objectives is heavily penalized by selecting a certain Pareto-optimal solution, then such solution will have a low likelihood of being recommended. Therefore, setting $\gamma = 0.25$ (as in Fig.s 3a, 3d, 3g, 3j) leads to excluding elements located in the external sections of the Pareto-front, because of their distance to the opposite extreme of the Pareto front.
   - with $\gamma$ larger than 0.5, the priority is given to achieving the greatest overall benefit, accepting the possibility of large penalties for one of the objectives. As a result, the set of recommended solutions (green triangles in Fig.3) can include the solution minimising the objective with the highest relative weight. For example, when $w_{SOC} = 0.8$ and $\gamma = 0.75$, the solution with minimum SOC is included in the set of Pareto-optimal solutions (Fig. 3l).
   - with $\gamma = 0.5$, the same relative importance is assigned to avoiding large penalties in one of the objectives, and to achieving the greatest overall benefit. This is the default value in our experiments, and all results presented in Section 3.2 were obtained setting $\gamma = 0.5$.

   This revised text aims at clarifying what the user should expect when changing the value of the group utility weight γ. In this passage, we also added references to the definition of γ in Sect. 2.3, and to the formula included in Appendix A.

   **References:**
   - Opricovic, S. and Tzeng, G.-H.: Compromise solution by MCDM methods: A comparative analysis of VIKOR and TOPSIS, European Journal of Operational Research, 156, 445–455, https://doi.org/10.1016/S0377-2217(03)00020-1, 2004